# ONE-SHOT CLUSTERING FOR CONTEXTUAL BANDITS WITH KNAPSACKS

## ABSTRACT

In this work, we study the problem of clustered linear contextual bandits with knapsack constraints, a setting that closely models real-world recommender systems. In such systems, the overwhelmed number of items makes it impractical to explore all options, and overexposing certain items can harm content diversity and fairness. To address these challenges, our algorithm clusters actions to enable knowledge transfer across similar items and incorporates global resource constraints to limit over-consumption. We provide a formal analysis showing that the algorithm achieves sublinear regret in the number of time periods, even without access to the full action set. Notably, we prove that it is sufficient to perform clustering once on a randomly selected subset of actions.

## 1 INTRODUCTION

In the contextual bandits problem (Langford & Zhang, 2007; Slivkins, 2011; Agarwal et al., 2014), an agent selects an action and collects a reward for that action over a sequence of rounds. At each round, the agent makes its choice based on the context for the current round and the feedback from previous rounds. The feedback only consists of the rewards for the chosen action, and the rewards of other actions remain unobserved. Online recommender systems (Tang et al., 2014) are often modeled as contextual bandit problems, where personalized recommendations are tailored for each user. The system receives feedback in the form of user interactions, such as clicks, likes, or comments, on the recommended item, but does not observe feedback for items that were not shown. Based on this partial feedback and the user's profile, the system continuously updates its policy to recommend new items.

Contextual bandits problem theoretically works in the recommender system. However, real world recommender system often suffers from:

- The constant introduction of new items makes the environment highly dynamic. Additionally, there is sparse user interaction when the item set is large. Contextual bandits must explore each item individually, leading to high sample complexity and cold-start issues as feedback per item remains scarce.

- Risk of depleting limited recommendation capacity. Contextual bandits do not account for constraints such as user attention span, content diversity quotas, or exposure fairness limits. Without modeling these factors, the system may over-recommend certain items, leading to user fatigue, reduced content diversity, or unfair exposure.

To solve the first challenge on how to present new items to users and use all available user-item preference information gathered, previous works (Yang et al., 2020a; Nguyen & Lauw, 2014; Gentile et al., 2017) have proposed clustered contextual bandits. It performs clustering over user interests, items, or user–item preferences, enabling clustered contextual bandits to transfer knowledge across similar users and improve learning efficiency. To address the second challenge of depleting limited recommendation capacity, prior research has extended the clustered bandit framework by incorporating global resource constraints. This method is known as bandit with knapsacks (Ma et al., 2024; Jiang & Ye, 2024; Slivkins et al., 2023; Li et al., 2021b; Han et al., 2023). To our knowledge, no prior work has combined these two ideas into a single algorithmic framework. Thus a natural question arises:

*Can we design a contextual-bandit algorithm that simultaneously exploits cluster structure and respects knapsack constraints, while retaining provable sublinear regret?*

In this work, we provide a positive answer to the above question. We introduce a new algorithm for clustered contextual bandits with knapsack constraints that achieves sublinear regret in the time horizon $T$. Arms are partitioned into unknown clusters. The arm's reward and $d$-dimension consumption are specified by a cluster-specific linear model in its observed context. At each round, the agent observes the i.i.d. context vector of each arm, select an arm, and get both a reward and a consumption vector. The goal of the algorithm is to maximize the sum of the rewards over periods. If the cumulative consumption of any resource exceeds its budget, the process terminates. Our model extends that of (Agrawal & Devanur, 2016) by considering clusters. The approach we employ builds on the algorithm of that paper. The main challenge lies in simultaneously learning the unknown cluster structure of the arms while guaranteeing that the average regret remains vanishing over time.

Our algorithm performs clustering only once. Initially, we sample a subset $\mathcal{S}$ of the total $K$ arms; any arm not in $\mathcal{S}$ is never played. The key intuition is that, to achieve vanishing average regret as the number of periods grows, we must access arms from every cluster and accurately estimate the fraction of the $K$ arms belonging to each cluster. Once $\mathcal{S}$ is fixed, we play each arm in $\mathcal{S}$ a predetermined number of times to gather sufficient observations for accurate clustering. Playing an arm yields $d + 1$ observations—one reward and $d$ resource-consumption values—each governed by a different linear model. For clustering, we adopt the classifier-Lasso method from econometrics (Su et al., 2016), which treats arm parameters as distinct (though not necessarily unequal) relative to cluster parameters. After clustering, the algorithm selects arms according to the optimism-in-the-face-of-uncertainty principle (Auer, 2002; Auer et al., 2002; Abbasi-Yadkori et al., 2011). In each period, we form optimistic estimates of both reward and resource consumption for every arm in $\mathcal{S}$, based on the observed context and the cluster-specific linear models. Since clustering may have some errors, some arms might be misgrouped and thus not share the true parameters of their assigned cluster. To facilitate regret analysis, we treat all arms assigned to a cluster as if they belonged to the same true cluster—but allow the linear-model error term to have a context-dependent expectation (with probability tied to the clustering error) instead of zero. In other words, we model imperfect clustering as measurement error in the context (Wansbeek & Meijer, 2001; Fuller, 2009). Finally, we leverage results from convex online learning to account for the depletion of each of the $d$ resources, following the approach in (Srebro et al., 2011) and the algorithm of (Agrawal & Devanur, 2016).

In regard to the number of time periods $T$, the regret of our algorithm increases at the rate $T^{1-\delta}$, for $\delta \in (0, \frac{1}{2})$, requiring the budget for each resource to be $B > \widetilde{O}(T^{2\delta})$. Thus, the regret is sublinear in $T$, and it can approach the square root rate as the resource budget approaches $T$. However, as it becomes clear in the next section, we must have $B < T$, since otherwise we could ignore the constraints. The main drawback of our algorithm is that its regret depends on the number of arms $K$, as it operates only on a subset of them. Nevertheless, our results suggest that such dependence may be inevitable. Overall, we believe that our approach points to a fruitful direction for the consideration of online learning problems that involve clusters and constraints.

**Roadmap.** The rest of the paper is organized as follows. In Section 2, we provide the related works on our algorithms. In Section 3, we provide notation and define the problem under consideration. In Section 4, we derive optimistic estimates of the reward and consumption parameters and establish confidence bounds for these estimate. Section 5 presents our `clusterLCBwK` algorithm along with its regret analysis. In Section 6, we discuss the implications of our results and potential directions for future work. Technical proofs and additional details are deferred to the appendix.

## 2 RELATED WORK

Some of the earliest works to consider the problem of bandits with knapsacks were (Badanidiyuru et al., 2014) and an earlier version of (Badanidiyuru et al., 2018), with both of them considering stochastic reward and resource consumption. The adversarial version of the problem was initially considered in (Immorlica et al., 2019), and consequently in (Kesselheim & Singla, 2020). An algorithm that achieves low expected regret for both the adversarial and the stochastic version is investigated in (Rangi et al., 2019), while in (Amani et al., 2019) bandit algorithms with constraints for safety-critical systems are proposed. General forms for the correlation of the reward and the resource

consumption are considered in (Cayci et al., 2020). (Devanur et al., 2011; Agrawal & Devanur, 2014; Agrawal et al., 2016) study computationally efficient algorithms for online learning problems with constraints, including bandits with knapsacks. (Agrawal & Devanur, 2016; Wu et al., 2015) consider the case where there is a single resource. From a result of (Dani et al., 2008), it is known that the regret of a linear contextual bandits algorithm must be at least linear in the dimensionality of the context. The topic of contextual bandits with behavioral constraints is presented in (Balakrishnan et al., 2018). (Yang et al., 2020b) studies contextual bandits with a resource constraint in recommender systems (Carlsson et al., 2021). Recent works focus on establishing high-probability performance guarantees (Ma et al., 2024; Deb et al., 2024; Jiang & Ye, 2024; Slivkins et al., 2023; Sivakumar et al., 2022), exploring symmetry properties (Li et al., 2021b), deriving optimal algorithms (Han et al., 2023), and under non-stationary settings (Liu et al., 2022; Lyu & Cheung, 2023; Zhang & Cheung, 2024).

In works that model recommender systems as bandits with clusters of users (Nguyen & Lauw, 2014; Gentile et al., 2014; 2017; Li & Zhang, 2018; Ban & He, 2021; Li et al., 2021a), the users arrive at the system exogenously. Besides the consideration of constraints and the fact that the model we study is not specific to an application, we cannot adopt techniques from the literature of recommender systems because we consider clusters of arms. Thus, the units to perform clustering on are endogenously and not exogenously provided, which makes our problem harder.

Clustering under mixed linear models (like ours) has attracted a lot of the interest in the literature (Zhong et al., 2016; Li & Liang, 2018; Chen et al., 2020; Kong et al., 2020; Chen et al., 2021; Pal et al., 2023; Wang et al., 2024; Cheng et al., 2023; Yang et al., 2024), but with the results of these works typically requiring a lower bound on the number of units to be clustered. Interestingly, similar versions of this problem have been considered in the literature of econometrics (Lin & Ng, 2012; Ando & Bai, 2016; Su et al., 2016; 2019; Gu & Volgushev, 2019; Okui & Wang, 2021). Here, we exploit results from the latter literature, since the assumptions they require are more appropriate to our case. In particular, we make assumptions about the joint rate of the number of sampled arms and of the observations collected for each such arm, instead of imposing a lower bound on the former.

## 3 PROBLEM DEFINITION AND NOTATION

In this section, we first introduce notation that we use throughout the paper, and then we provide the definition of the problem we consider. For any positive integer $N$, we use $[N]$ to denote the set $\{1, 2, \cdots, N\}$. We use $\mathbb{E}[X]$ to denote the expectation of the random variable $X$, and $\Pr[E]$ to denote the probability of the event $E$. The notation $\mathbb{1}[f]$ is used for the indicator variable which outputs 1 if $f$ holds and 0 otherwise. Boldface lower case letters are reserved for vectors, and boldface upper case letters for matrices. The vector $\mathbf{1}_d$ denotes the $d$-dimensional vector that has the value 1 in every dimension, and $\mathbf{0}_d$ is defined equivalently. Calligraphic upper case letters, e.g., $\mathcal{S}$, denote sets. The probability simplex in $d$ dimensions is denoted as $\Delta^d$. For a square matrix $\boldsymbol{A}$, we define the matrix norm of the vector $\boldsymbol{x}$ as $\|\boldsymbol{x}\|_{\boldsymbol{A}} = \sqrt{\boldsymbol{x}^\top \boldsymbol{A} \boldsymbol{x}}$.

The total number of arms is denoted by $K$. A subset of arms $\mathcal{S} \subset [K]$ is sampled from $[K]$. Arms not in $\mathcal{S}$ never being played. The number of clusters by $C \in \mathbb{N}$. Each arm belongs to one cluster, with $c(a) \in [C]$ denoting the cluster of arm $a \in [K]$. The algorithm is given a budget $B \in \mathbb{R}_+$ and the number of clusters $C$, but not the arm membership in the clusters. In each period $t \in [T]$, the algorithm observes the context of all the arms $X_t = [x_t(1), x_t(2), \ldots, x_t(K)] \in [0, 1]^{m \times K}$. $\boldsymbol{x}_t(a)$ is the context vector of arm $a$ in time $t$. $m \in \mathbb{N}$ is the dimension of the context vector. Then, the agent chooses arm $a_t \in [K]$, and finally observes reward $r_t(a_t) \in [0, 1]$ and consumption vector $\boldsymbol{v}_t(a_t) \in [0, 1]^d$. The algorithm can also play the "no-op" action that deterministically gives zero reward and consumption. The goal of the algorithm is to maximize the total reward $\sum_{t \in [T]} r_t(a_t)$ under the constraints $\sum_{t \in [T]} \boldsymbol{v}_t(a_t) \leqslant B \cdot \mathbf{1}_d$. Here, $\boldsymbol{v}_t(a_t)$ is a $d$-dimensional vector indicating how much of each resource is consumed by choosing arm $a_t$ at time $t$. Notice that it is without loss of generality to consider the same budget for each of the $d$ resources, since this can imposed by normalization.

Moreover, we assume that the reward and the consumption are generated by cluster-specific linear models, as indicated by the following assumptions.

**Assumption 3.1** (Linearity). *For each cluster $c \in [C]$, there is an unknown parameter vector $\boldsymbol{\mu}_c \in [0,1]^m$ and an unknown parameter matrix $\boldsymbol{W}_c \in [0,1]^{m \times d}$ such that for all rounds $t \in [T]$ and all arms $a \in [K]$,*

$$r_t(a) = \boldsymbol{\mu}_{c(a)}^\top \boldsymbol{x}_t(a) + g_t(a) \tag{1}$$

$$\boldsymbol{v}_t(a) = \boldsymbol{W}_{c(a)}^\top \boldsymbol{x}_t(a) + \boldsymbol{q}_t(a) \tag{2}$$

*where $\boldsymbol{\mu}_{c(a)} = \boldsymbol{\mu}_c$ and $\boldsymbol{W}_{c(a)} = \boldsymbol{W}_c$ if $c(a) = c$, and $g_t(a) \in \mathbb{R}$ and $\boldsymbol{q}_t(a) \in \mathbb{R}^d$ are additive error terms.*

We make the following assumptions about the error terms.

**Assumption 3.2** (Zero conditional mean). *We assume $\mathbb{E}[g_t(a)|\boldsymbol{x}_t(a)] = 0$, and $\mathbb{E}[\boldsymbol{q}_t(a)|\boldsymbol{x}_t(a)] = \boldsymbol{0}_d$.*

**Assumption 3.3.** *We assume $|g_t(a)| \leqslant 2R$, and $\|\boldsymbol{q}_t(a)\|_\infty \leqslant 2R$.*

Furthermore, we make the following i.i.d. assumption about the context.

**Assumption 3.4** (i.i.d.). *The context $\boldsymbol{x}_t(a)$ is i.i.d. across arms and periods.*

In the algorithm also reply on Assumption A.5, Assumption A.6, Assumption A.7, Assumption A.8. For brevity, we direct reader to Appendix A for more details.

We let the vector $\boldsymbol{p} = (p_1, \ldots, p_C) \in \Delta^C$ denote the proportions of arms that are in each cluster, i.e., $p_c = \frac{1}{K} \sum_{a \in [K]} \mathbb{1}[c(a) = c]$ for $c \in [C]$. The smallest proportion in $\boldsymbol{p}$ is denoted as $p_{\min} := \min_{c \in [C]} p_c$. Even though the algorithm does not know the other proportions in $\boldsymbol{p}$, some knowledge about $p_{\min}$ is required for the clustering, as described later. Also, notice that $p_{\min} \leqslant 1/C$ by definition.

### 3.1 BENCHMARK

Our goal is to devise an algorithm that achieves sublinear regret in the number of time periods $T$. We employ as benchmark the expected reward of the optimal static policy which needs to satisfy the consumption constraints only in expectation. This benchmark policy knows the reward and consumption parameters for each cluster, as well as the cluster membership of each arm. It is known (Devanur et al., 2011; Agrawal & Devanur, 2016; Badanidiyuru et al., 2018) that the optimal static policy achieves the same expected reward to the optimal adaptive policy which knows the distribution of the context and needs to satisfy the constraints for the realizations of the resource consumption.

**Definition 3.5** (Optimal Static Policy). *Let $\boldsymbol{X} = \big(\boldsymbol{x}(1), \ldots, \boldsymbol{x}(K)\big) \in [0,1]^{m \times K}$ be the matrix with the context of all arms in an arbitrary period, and $\pi(i, \boldsymbol{X}) \in [0,1]$ the probability with which the action $i \in [K] \cup \{\text{"no-op"}\}$ is taken by the static policy $\pi$ when the context is $\boldsymbol{X}$. The per-period expected reward and consumption vector of $\pi$ are respectively defined as*

$$r(\pi) := \mathop{\mathbb{E}}_{\boldsymbol{X}} \Big[ \sum_{a \in [K]} \boldsymbol{\mu}_{c(a)}^\top \boldsymbol{x}(a) \pi(a, \boldsymbol{X}) \Big],$$

$$\boldsymbol{v}(\pi) := \mathop{\mathbb{E}}_{\boldsymbol{X}} \Big[ \sum_{a \in [K]} \boldsymbol{W}_{c(a)}^\top \boldsymbol{x}(a) \pi(a, \boldsymbol{X}) \Big]. \tag{3}$$

*With $\Pi$ denoting the set of all static policies, the optimal static policy is defined as*

$$\pi^* := \arg \max_{\pi \in \Pi} r(\pi) \text{ subject to } \boldsymbol{v}(\pi) \leqslant \frac{B}{T} \cdot \boldsymbol{1}_d.$$

*The expected total reward of $\pi^*$ is defined as*

$$\text{OPT} := T \cdot r(\pi^*).$$

Since the "no-op" action is allowed, the policy $\pi^*$ is feasible and the definition of OPT is valid. Having specified the benchmark, we can now define the regret of our algorithm.

**Definition 3.6** (Regret). *The regret of the algorithm for $T$ time periods is defined as*

$$\text{regret}(T) := \text{OPT} - \sum_{t=1}^{T} r_t(a_t). \tag{4}$$

## 4 Optimism in the Face of Uncertainty

In this section, we describe how optimistic estimates of the reward and consumption parameters are derived in time periods after the clustering is performed. Such estimates will consequently be utilized to devise a no-regret algorithm in the next section. At period $t > N_{\mathcal{S}} \cdot T_0$, the parameters of cluster $c \in [C]$ are estimated using all the observations prior to $t$ that involve arms that have been assigned to $c$. We denote by $t_c < t$ the number of periods in which arms estimated to be in $c$ have been played prior to period $t$, i.e., $t_c := \sum_{i=1}^{t-1} \mathbb{1}[\hat{c}(a_i) = c]$.

To formalize the estimation error that arises both from inherent stochasticity and from occasional clustering mistakes, we begin by introducing a general noisy-observation model. Specifically, we view each observed outcome as the true linear predictor plus an additive noise term that decomposes into a zero–mean fluctuation and a bias component due to mis-clustering.

**Definition 4.1.** *At time step $t \in [T]$, let $\boldsymbol{x}_t \in [0,1]^m$ denote the observed context vector, and let $y_t \in [0,1]$ be the corresponding observed outcome. We define $y_t := \boldsymbol{\mu}^\top \boldsymbol{x}_t + \eta_t$, where $\boldsymbol{\mu} \in [0,1]^m$ is an unknown parameter vector, and $\eta_t$ is an error term composed of two components: $\eta_t = u_t + h_t$, with $\mathbb{E}[u_t \mid \boldsymbol{x}_t] = 0$ (from Assumption 3.2) and $|u_t| \leqslant 2R$ (from Assumption 3.3). The second component $h_t$ accounts for clustering mismatch and is defined as*

$$h_t = \begin{cases} 0 & \text{w.p. } 1 - \epsilon, \\ \boldsymbol{\gamma}^\top \boldsymbol{x}_t & \text{w.p. } \epsilon, \end{cases}$$

*where $\boldsymbol{\gamma} \in [-1,1]^m$ is perceived as the element-wise difference between parameters of different clusters.*

Building on the noisy observation model of Definition 4.1, we now turn to estimating the parameter vector $\boldsymbol{\mu}$ via regularized least squares. At each time step $t$, all past context–outcome pairs $\{(\boldsymbol{x}_i, y_i)\}_{i=1}^{t-1}$

are aggregated into a design matrix, and a ridge regression estimator is computed to balance data fitting against overfitting. Concretely, we form the regularized covariance matrix and then solve for $\hat{\boldsymbol{\mu}}_t$ as follows.

**Definition 4.2.** *Let $\lambda_2 > 0$ be the regularization parameter. At each time step $i < t$, the agent observes a context vector $\boldsymbol{x}_i \in [0,1]^m$ and its corresponding scalar outcome $y_i \in [0,1]$, where $y_i = \boldsymbol{\mu}^\top \boldsymbol{x}_i + \eta_i$. Then the regularized matrix $\boldsymbol{M}_t$ and the regression estimator $\hat{\boldsymbol{\mu}}_t$ at time $t$ are defined as:*

$$\boldsymbol{M}_t := \lambda_2 \boldsymbol{I}_m + \sum_{i=1}^{t-1} \boldsymbol{x}_i \boldsymbol{x}_i^\top,$$

$$\hat{\boldsymbol{\mu}}_t := \boldsymbol{M}_t^{-1} \sum_{i=1}^{t-1} \boldsymbol{x}_i y_i. \tag{5}$$

By substituting $r_t(a)$, $\boldsymbol{\mu}_c$ and $\epsilon_c$ for $y_t$, $\boldsymbol{\mu}$ and $\epsilon$ in Definition 4.1, where $c = \hat{c}(a)$ for each $a \in \mathcal{S}$, the concentration results of Section A.4 apply to the reward parameter of cluster $c$ using its $t_c$ observations. Similarly, letting $\boldsymbol{w}_{c,j}$ be the $j$th column of $\boldsymbol{W}_c$ for $j \in [d]$, the same bounds hold for each consumption dimension. In particular, with $R = \frac{1}{2}$ and $\lambda_2 = 1$, we define the cluster-specific design matrix

$$\boldsymbol{M}_{c,t} := \boldsymbol{I}_m + \sum_{i<t:\hat{c}(a_i)=c} \boldsymbol{x}_i(a_i)\boldsymbol{x}_i(a_i)^\top.$$

The parameters of cluster $c \in [C]$ are estimated at period $t$ as

$$\hat{\boldsymbol{\mu}}_{c,t} := \boldsymbol{M}_{c,t}^{-1} \sum_{i<t:\hat{c}(a_i)=c} \boldsymbol{x}_i(a_i) r_i(a_i)$$

$$\widehat{\boldsymbol{W}}_{c,t} := \boldsymbol{M}_{c,t}^{-1} \sum_{i<t:\hat{c}(a_i)=c} \boldsymbol{x}_i(a_i) \boldsymbol{v}_i(a_i)^\top.$$

**Definition 4.3.** *Let* $\zeta \in (0, 1)$ *be a confidence parameter. At time step* $t$, *define the confidence ellipsoid* $\mathcal{C}_t \subset \mathbb{R}^m$ *as the set of vectors* $\boldsymbol{\beta}$ *satisfying*

$$\mathcal{C}_t := \left\{ \boldsymbol{\beta} \in \mathbb{R}^m : \|\boldsymbol{\beta} - \widehat{\boldsymbol{\mu}}_t\|_{\boldsymbol{M}_t} \leqslant \rho_t \right\},$$

*where the radius* $\rho_t$ *is given by*

$$\rho_t := 2(R+1)\sqrt{m \log\left(\frac{tm}{\lambda_2 \zeta}\right)} + \epsilon m \sqrt{t} + \sqrt{\lambda_2 m}.$$

For brevity, We refer the reader to Section A.4 for additional details.

With $R = \frac{1}{2}$ and $\lambda_2 = 1$, , we can define confidence ellipsoids for the parameters of each cluster and derive optimistic estimates. The confidence ellipsoid of the reward vector of cluster $c$ at period $t$ is defined as $\mathcal{C}_{\mu,c,t} := \{\boldsymbol{\beta} \in \mathbb{R}^m : \|\boldsymbol{\beta} - \widehat{\boldsymbol{\mu}}_{c,t}\|_{\boldsymbol{M}_{c,t}} \leqslant 3\sqrt{m \log(t_c m/\zeta)} + \epsilon m \sqrt{t_c} + \sqrt{m}\}$, and the optimistic estimate of the reward parameter for arm $a \in S$ at period $t$ is defined as

$$\widetilde{\boldsymbol{\mu}}_{a,t} := \arg \max_{\boldsymbol{\beta} \in \mathcal{C}_{\mu,\widehat{c}(a),t}} \boldsymbol{x}_t(a)^\top \boldsymbol{\beta}. \tag{6}$$

We can similarly define the confidence ellipsoid for the vector of the consumption dimension $j \in [d]$ for cluster $c$ at period $t$ as

$$\mathcal{C}_{w,c,t,j} := \left\{ \boldsymbol{\beta} \in \mathbb{R}^m : \|\boldsymbol{\beta} - \widehat{\boldsymbol{w}}_{c,t,j}\|_{\boldsymbol{M}_{c,t}} \leqslant 3\sqrt{m \log(dt_c m/\zeta)} + \epsilon m \sqrt{t_c} + \sqrt{m} \right\},$$

where $\widehat{\boldsymbol{w}}_{c,t,j}$ is the $j^{\text{th}}$ column of $\widehat{\boldsymbol{W}}_{c,t}$. Given a vector $\boldsymbol{\theta}_t \in [0,1]^d$, we define the optimistic consumption estimate for arm $a \in \mathcal{S}$ at time $t$ by choosing the matrix in the Cartesian product of the $d$ confidence sets that minimizes the weighted consumption:

$$\widetilde{\boldsymbol{W}}_{a,t} := \arg \min_{\boldsymbol{W} \in \times_{j=1}^d \mathcal{C}_{w,\widehat{c}(a),t,j}} \boldsymbol{x}_t(a)^\top \boldsymbol{W} \boldsymbol{\theta}_t. \tag{7}$$

By Lemma A.11 and the union bound, $\boldsymbol{W}_c$ is in $\times_{j=1}^d \mathcal{C}_{w,c,t,j}$ with probability at least $1 - \zeta$. The vector $\boldsymbol{\theta}_t$ will allow the algorithm in the next section to translate consumption into reward, so that arms that are expected to consume plenty of scarce resources will be relatively less appealing. For the optimistic estimate of a reward vector, the maximizer is picked in Eq. (6) since we want the choice of an arm to result in high reward. However, since large budget losses are undesirable, the optimistic estimate of a consumption matrix is taken to be the minimizer in Eq. (7). The following lemma relates the optimistic estimates to the true parameters.

**Lemma 4.4** (Informal version of Lemma C.1). *Given clustering* $\{\widehat{c}(a)\}_{a \in \mathcal{S}}$ *and vectors* $\{\boldsymbol{\theta}_i\}_{i=N_{\mathcal{S}} \cdot T_0 + 1}^t$, *where* $\boldsymbol{\theta}_i \in [0,1]^d$, *with probability at least* $1 - \zeta$ *we have that for any* $a \in \mathcal{S}$,

    *a)* $\boldsymbol{x}_t(a)^\top \big(\widetilde{\boldsymbol{\mu}}_{a,t} - \boldsymbol{\mu}_{\widehat{c}(a)}\big) \geqslant 0$,

    *b)* $\boldsymbol{x}_t(a)^\top \big(\widetilde{\boldsymbol{W}}_{a,t} - \boldsymbol{W}_{\widehat{c}(a)}\big)\boldsymbol{\theta}_t \leqslant 0$,

    *c)* $|\sum_{i=N_{\mathcal{S}} \cdot T_0 + 1}^t \boldsymbol{x}_i(a_i)^\top (\widetilde{\boldsymbol{\mu}}_{a_i,i} - \boldsymbol{\mu}_{\widehat{c}(a_i)})| \leqslant \rho$,

    *d)* $\|\sum_{i=N_{\mathcal{S}} \cdot T_0 + 1}^t \boldsymbol{x}_i(a_i)^\top (\widetilde{\boldsymbol{W}}_{a_i,i} - \boldsymbol{W}_{\widehat{c}(a_i)})\|_\infty \leqslant \rho$,

*where* $\rho$ *is given by:*

$$\rho := 4Cm\sqrt{t \log(tm/\zeta) \log(t)} + \epsilon_c m^{\frac{3}{2}} t \sqrt{\log(t)}.$$

For brevity, the proof of Lemma 4.4 is deferred to Section C.

## 5 ALGORITHM

In this section, we present our algorithm for the problem of clustered linear contextual bandits with knapsacks (clusterLCBwK) and results related to its regret. Initially, clustering is performed as specified in subsection A.2. We show that a small, randomly sampled subset of arms is sufficient to identify all underlying clusters with high probability.

**Lemma 5.1** (informal version of Lemma A.4). *For parameter $\delta > 0$, if the set $\mathcal{S}$ is formed by sampling $N_{\mathcal{S}} = O\big(p_{\min}^{-1}(T^{\delta} + \log C)\big)$ arms, where each arm is sampled with equal probability, then $\mathcal{S}$ covers the clusters, i.e., $\cup_{a \in S}\{c(a)\} = [C]$, with probability at least $1 - O(T^{-2\delta})$.*

**Lemma 5.2** (Informal version of Lemma D.1). *If $\max_{\pi \in \{\pi^*, \pi'\}} |r(\pi) - \rho(\pi)| \leqslant \epsilon'$ then $|r(\pi^*) - \rho(\pi')| \leqslant \epsilon'$.*

With Lemma 5.2, it suffices to prove the difference $\max_{\pi \in \{\pi^*, \pi'\}} |r(\pi) - \rho(\pi)|$ is small. We first consider $\pi^*$ and then $\pi'$.

**Lemma 5.3** (Informal version of Lemma D.2). *$|r(\pi^*) - \rho(\pi^*)| < o(1)$.*

**Lemma 5.4** (Informal version of Lemma D.3). *$|r(\pi') - \rho(\pi')| < o(1)$.*

Combining Lemmas 5.3 and 5.4, we conclude that $\max_{\pi \in \{\pi^*, \pi'\}} |r(\pi) - \rho(\pi)|$ is vanishing. It remains to show that our empirical estimate of the optimal reward is accurate. Using the initial $N_{\mathcal{S}}T_0$ samples, we estimate the reward of the optimal static policy, as formalized in the following lemma.

**Lemma 5.5.** *Let $\widehat{\mathrm{OPT}} := T \cdot \hat{r}$, where $\hat{r}$ is the estimate of $r(\pi^*)$ based on the initial $\mathcal{N}_{\mathcal{S}}T_0$ random samples, defined as*

$$\hat{r} := \max_{\pi} \frac{K}{N_{\mathcal{S}}^2 T_0} \sum_{t=1}^{N_{\mathcal{S}}T_0} \sum_{a \in \mathcal{S}} \widehat{\boldsymbol{\mu}}_{\hat{c}(a), N_{\mathcal{S}}T_0+1}^{\top} \boldsymbol{x}_t(a)\pi(a, \boldsymbol{X}_t)$$

$$\text{s.t.} \quad \frac{K}{N_{\mathcal{S}}^2 T_0} \sum_{t=1}^{N_{\mathcal{S}}T_0} \sum_{a \in \mathcal{S}} \widehat{\boldsymbol{W}}_{\hat{c}(a), N_{\mathcal{S}}T_0+1}^{\top} \boldsymbol{x}_t(a)\pi(a, \boldsymbol{X}_t) \leqslant \frac{B}{T}\mathbf{1}_d.$$

*Then, $\widehat{\mathrm{OPT}} - \mathrm{OPT} = o(1)$ with high probability.*

*Proof.* For notational convenience, let

$$\rho(\pi) := \frac{K}{N_{\mathcal{S}}^2 T_0} \sum_{t=1}^{N_{\mathcal{S}}T_0} \sum_{a \in \mathcal{S}} \widehat{\boldsymbol{\mu}}_{\hat{c}(a), N_{\mathcal{S}}T_0+1}^{\top} \boldsymbol{x}_t(a)\pi(a, \boldsymbol{X}_t).$$

Denote the maximizer of the program of the lemma by $\pi'$, i.e.,

$$\pi' := \arg \max_{\pi} \rho(\pi) \text{ s.t. } \frac{K}{N_{\mathcal{S}}^2 T_0} \sum_{t=1}^{N_{\mathcal{S}}T_0} \sum_{a \in \mathcal{S}} \widehat{\boldsymbol{W}}_{\hat{c}(a), N_{\mathcal{S}}T_0+1}^{\top} \boldsymbol{x}_t(a)\pi(a, \boldsymbol{X}_t) \leqslant \frac{B}{T}\mathbf{1}_d.$$

By the definition of the benchmark policy in Section 3, we have $\pi^* = \arg \max_{\pi \in \Pi} r(\pi)$ subject to $\boldsymbol{v}(\pi) \leqslant \frac{B}{T} \cdot \mathbf{1}_d$. For ease of exposition, we will first ignore the constraints and account for them later. Notice that we want to prove that the difference $|r(\pi^*) - \rho(\pi')|$ is small. The next lemma allows us to work with a more convenient term instead. $\square$

The estimate $\widehat{\mathrm{OPT}}$ is then used to define the variable $Z$ which will contribute to the choice made by the algorithm in periods $t > N_{\mathcal{S}}T_0$ by appropriately weighting the optimistic estimates of the consumption. In particular, this variable is defined as

$$Z := \frac{N_{\mathcal{S}}\widehat{\mathrm{OPT}}}{2KB'}, \tag{8}$$

where $B' := B - N_{\mathcal{S}}T_0$ and we also define $T'$ similarly, i.e., $T' := T - N_{\mathcal{S}}T_0$. In periods $t > N_{\mathcal{S}}T_0$, the choice of the algorithm is

$$a_t = \arg \max_{a \in \mathcal{S}} \boldsymbol{x}_t(a)^{\top} (\widetilde{\boldsymbol{\mu}}_{a,t} - Z\widehat{\boldsymbol{W}}_{a,t}\boldsymbol{\theta}_t),$$

where $\boldsymbol{\theta}_t \in [0,1]^d$ is the choice of the online mirror descent algorithm when at the previous period the payoff $\boldsymbol{\theta}_{t-1}^{\top}(\boldsymbol{v}(a_{t-1}) - \frac{B'}{T'}\mathbf{1}_d)$ is observed. Thus, in effect, $Z$ and $\boldsymbol{\theta}_t$ allow the resource consumption to be compared to the reward, so that arms estimated to consume a lot of a scarce resource can be avoided.

Our algorithm is an extension of the linear contextual bandits with knapsacks (linCBwK) algorithm (Agrawal & Devanur, 2016), but with the clustering step having been incorporated and accounted for in the derivation of the regret. Moreover, our estimation of OPT utilizes the initial randomly collected $N_S T_0$ samples, and is different to the corresponding estimation in linCBwK which would lead to additional sampling and thus higher regret in our case.

---

**Algorithm 1** clusterLCBwK

---

1: $N_S \leftarrow O\big(p_{\min}^{-1}(T^\delta + \log C)\big)$
2: $S \leftarrow$ random subset of $[K]$ with size $N_S$
3: $T_0 \leftarrow N_S$
4: $\forall a \in S$, play $T_0$ times the arm $a$
5: Cluster the arms in $S$ per Eq. (11)
6: Compute $Z$ per Eq. (8)
7: **for** $t = N_S T_0 + 1, \ldots, T$ **do**
8:      $\forall a \in S$, obtain $\widetilde{\boldsymbol{\mu}}_{a,t}$ and $\widehat{\boldsymbol{W}}_{a,t}$ per Eq. (6), (7)
9:      $a_t \leftarrow \arg\max_{a \in S} \boldsymbol{x}_t(a)^\top (\widetilde{\boldsymbol{\mu}}_{a,t} - Z\widetilde{\boldsymbol{W}}_{a,t}\boldsymbol{\theta}_t)$
10:      Play the arm $a_t$, and observe $r_t(a_t)$ and $\boldsymbol{v}_t(a_t)$
11:      **if** $\exists j \in [d] : \sum_{i=1}^{t} \boldsymbol{v}_i(a_i)^\intercal \boldsymbol{e}_j \geqslant B$ **then** exit
12:      Update $\boldsymbol{M}_{c,t+1}$, $\widehat{\boldsymbol{\mu}}_{c,t+1}$, and $\widehat{\boldsymbol{W}}_{c,t+1}$, for $c = \widehat{c}(a_t)$
13:      Update $\boldsymbol{\theta}_{t+1}$ with the online mirror descent algorithm for payoff $\boldsymbol{\theta}_t^\top (\boldsymbol{v}_t(a_t) - \frac{B'}{T'}\boldsymbol{1}_d)$
14: **end for**

---

**Lemma 5.6** (Informal version of Lemma D.4). *With probability at least $(1 - \zeta)^3$ we have:*

*a)* $|\sum_{t=N_S T_0 + 1}^{T_\omega} r_t(a_t) - \boldsymbol{x}_t(a_t)^\top \widetilde{\boldsymbol{\mu}}_{a_t,t}| \leqslant R(T),$

*b)* $\|\sum_{t=N_S T_0 + 1}^{T_\omega} \boldsymbol{v}_t(a_t) - \boldsymbol{x}_t(a_t)^\top \widetilde{\boldsymbol{W}}_{a_t,t}\|_\infty \leqslant R(T).$

Now, let $S_1$ denote the subset of $S$ that contains correctly clustered arms, $S_1 := \{a \in S : \widehat{c}(a) = c(a)\}$.

The following lemma provides a lower bound related to the choice of the algorithm.

**Lemma 5.7** (Informal version of Lemma D.5). *For $t > N_S T_0$, the following inequality holds with high probability:*

$$\boldsymbol{x}_t(a_t)^\top (\widetilde{\boldsymbol{\mu}}_{a_t,t} - Z\widetilde{\boldsymbol{W}}_{a_t,t}\boldsymbol{\theta}_t) \geqslant \frac{1}{\sum_{a' \in S_1} \pi^*(a', \boldsymbol{X}_t)} \sum_{a \in S_1} \pi^*(a, \boldsymbol{X}_t) \cdot \boldsymbol{x}_t(a)^\top \big(\boldsymbol{\mu}_{c(a)} - ZW_{c(a)}\boldsymbol{\theta}_t\big).$$

Lemma 5.7 implies the weaker condition where expectation is taken over $\boldsymbol{X_t}$ only and it is conditional on the past realizations of the context.

**Lemma 5.8** (Informal version of Lemma D.6). *The following inequality holds with high probability:*

$$\sum_{t=N_S T_0 + 1}^{T_\omega} \mathop{\mathbb{E}}_{\boldsymbol{X}_t} \Big[ \boldsymbol{x}_t(a_t)^\top \widetilde{\boldsymbol{\mu}}_{a_t,t} \Big] \geqslant O\Big(\frac{N_S \cdot T_\omega}{K \cdot T} \text{OPT}$$

$$+ Z \sum_{t=N_S T_0 + 1}^{T_\omega} \mathop{\mathbb{E}}_{\boldsymbol{X}_t} \Big[ \boldsymbol{x}_t(a_t)^\top \widetilde{\boldsymbol{W}}_{a_t,t} - \frac{N_S}{K} \cdot \frac{B}{T}\boldsymbol{1}_d \Big]\boldsymbol{\theta}_t\Big).$$

Since in the first $N_S T_0$ periods the choices are made randomly, and so $N_S T_0$ of the budget can potentially be consumed, the following lemma which is known from the literature is expressed in terms of $B' = B - N_S T_0$ and $T' = T - N_S T_0$ rather than $B$ and $T$.

**Lemma 5.9** (Lemma 9 of (Agrawal & Devanur, 2016)).

$$\sum_{t=N_ST_0+1}^{T_\omega} \Big(\boldsymbol{x}_t(a_t)^\top \widetilde{\boldsymbol{W}}_{a_t,t} - \frac{B'}{T'}\boldsymbol{1}_d\Big)\boldsymbol{\theta}_t \geqslant B'\Big(1 - \frac{T_\omega - N_ST_0}{T'}\Big) - R(T).$$

The following theorem is our main result, showing that the regret of Algorithm 1 is sublinear in $T$.

**Theorem 5.10** (Main Result, informal version of D.8). *For $\delta \in (0, \frac{1}{2})$, and $B > N_ST_0$, with high probability*

$$\mathrm{regret}(T) \leqslant O\Big(R(T)\big(1 + \frac{N_S \, \mathrm{OPT}}{KB'}\big) + \mathrm{OPT}\big(1 - \frac{N_S}{K}\big)\Big)$$

*where*

$$R(T) = O\Big(Cp_{\min}^{-1}m^{\frac{3}{2}}T^{1-\delta}\sqrt{\log(T)}\Big).$$

Since $N_S/K \leqslant 1$ and the exponent of $T$ is at most $1 - \delta < 1$, the regret is sublinear in the number of time periods. In the special case where we sampled all arms, i.e., $N_S = K$, the bound simplifies to $O\big(R(T)(1 + \mathrm{OPT}/B')\big)$ where $B' = B - N_ST_0$. This matches the structure of non-clustered knapsacks bound $O\big(R'(T)(1 + \mathrm{OPT}/B)\big)$ (Agrawal & Devanur, 2016), with $R'(T) = \widetilde{O}(mT^{\frac{1}{2}})$ Moreover, the difference between $B' = B - N_ST_0$ and $B$ in the division of OPT reflects the loss due to the initial $N_ST_0$ periods of obtaining samples for the clustering.

The probability $1 - \zeta$ that appears in Lemma 4.4 affects our regret only through $R'(T)$. Considering the case that is of greater interest here, where a subset of the arms is sampled, i.e., $N_S < K$, as $N_S$ decreases, the regret due to $R(T)$ decreases until $\frac{N_S \, \mathrm{OPT}}{KB'} < 1$ ,while the term $\mathrm{OPT}\big(1 - \frac{N_S}{K}\big)$ increases. These two terms express two different sources of regret. More specifically, $R(T)$ decreases in the number of sampled arms because it captures the difference in the performance between our algorithm and the optimal static policy with respect to the portion of the arms that are sampled. The second term, $\mathrm{OPT}\big(1 - \frac{N_S}{K}\big)$, reflects the loss suffered due to the fact that as the number of sampled arms decreases, the options that are available to our algorithm become fewer. Thus, the algorithm will sometimes miss the opportunity to pick arms with favorable context due to such arms having been discarded. Therefore, the choice of $N_S$ determines the impact of each of these two sources of regret. Understanding the consequences of this choice can be especially important in applications where for practical reasons operating on the set $[K]$ is infeasible. Despite that our results suggest that the dependence of the regret on the number of arms $K$ may be inevitable, the weakening of this dependence is an open question.

## 6 DISCUSSION

We have shown that regret sublinear in the number of time periods can be achieved for the problem of linear contextual bandits with knapsacks and clusters of arms. Our approach does not require access to all the available arms, and can thus be utilized in applications where the heterogeneity of the available choices can be meaningfully summarized with clusters, and where individually considering each possible choice is unrealistic, e.g., in online advertising campaigns. It is among our beliefs that the study of forms for summarizing choice heterogeneity in online learning can be a fruitful research direction.

Furthermore, our approach can be extended in a number of ways. For instance, the initially sampled arms can serve only as a basis for the clustering, and with additional arms being explored and clustered in later steps of the algorithm. Also, the results in (Su et al., 2016) suggest that the assumption about knowing the number of the clusters can be relaxed. The improvement of the dependency of the regret on the total number of arms $K$ is a question we find interesting to be explored. Notice that we could have improved this dependency here by allowing the number of sampled arms $N_S$ to be a function of $K$. However, the budget constraints would then depend on $K$.

## ETHIC STATEMENT

This paper does not involve human subjects, personally identifiable data, or sensitive applications. We do not foresee direct ethical risks. We follow the ICLR Code of Ethics and affirm that all aspects of this research comply with the principles of fairness, transparency, and integrity.

## REPRODUCIBILITY STATEMENT

We ensure reproducibility of our theoretical results by including all formal assumptions, definitions, and complete proofs in the appendix. The main text states each theorem clearly and refers to the detailed proofs. No external data or software is required.

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

# Appendix

## Appendix

**Roadmap.** The appendix is organized as follows.

- In Section A we provide the preliminary of this work.
- In Section B, we present supplementary proofs for Section A.
- In Section C, we present supplementary Lemmas for Section 4.
- In Section D we present the proofs of the results from Section 5 related to the regret of the algorithm.

## A    PRELIMINARIES

In this section we present results and technical ingredients that will allow us to derive the regret.

### A.1    PROBABILITY TOOLS

**Lemma A.1** (Cantelli's inequality). *For random variable $X$ and constant $\xi > 0$,*

$$\Pr\left[X \geqslant \mathbb{E}[X] + \xi\right] \leqslant \frac{Var[X]}{Var[X] + \xi^2}$$

**Lemma A.2** (Azuma-Hoeffding inequality). *For a martingale $\{X_i\}_{i=0,1,\ldots}$ and constant $\xi > 0$, if $|X_i - X_{i-1}| \leqslant c_i$ almost surely, then*

$$\Pr\left[|X_N - X_0| \geqslant \xi\right] \leqslant 2 \exp\left(-\frac{\xi^2}{2\sum_{i=1}^{N} c_i^2}\right)$$

**Lemma A.3** ((Su et al., 2016), pp. 2250). $\Pr[\widehat{c}(a) = c(a)] = 1 - o(N_{\mathcal{S}}^{-1})$

### A.2    CLUSTERING UNDER A MIXED LINEAR MODEL

In order to devise an algorithm that achieves low regret, it will be necessary to learn the parameters $\boldsymbol{\mu}_c$ and $\boldsymbol{W}_c$ of each cluster $c \in [C]$. Consequently, we need to assign arms to clusters, with the correctness of an arm's assignment being defined only up to permutation of the cluster identities, as a cluster is defined by its members. However, clustering all of the arms requires obtaining at least $K$ samples, which could be undesirable. Instead, our approach allows clustering and operating on a subset of the arms.

While clustering will be performed only for a subset of the $K$ arms, it is important that this subset covers all the $C$ clusters. We initially choose (without playing) at random a subset of the $K$ arms, denoted by $\mathcal{S} \subseteq [K]$, with $|\mathcal{S}| = N_{\mathcal{S}}$. The set of arms $[K]\backslash\mathcal{S}$ that is not chosen will be discarded for all the $T$ time periods. Of course, the regret will be later derived given that the benchmark static policy has all the $K$ arms in its availability. The following result indicates the size of the set $\mathcal{S}$ needed in order to cover all of the clusters with a given probability. We use $p_{\min}$ to denote the proportion of arms belonging to the smallest cluster.

**Lemma A.4** (formal version of Lemma 5.1). *For parameter $\delta > 0$, if the set $\mathcal{S}$ is formed by sampling $N_{\mathcal{S}} = O\big(p_{\min}^{-1}(T^\delta + \log C)\big)$ arms, where each arm is sampled with equal probability, then $\mathcal{S}$ covers the clusters, i.e., $\cup_{a \in S}\{c(a)\} = [C]$, with probability at least $1 - O(T^{-2\delta})$.*

*Proof.* In this proof, we will treat the portions in $\boldsymbol{p}$ as probabilities from a distribution by sampling an arm with replacement, instead of without replacement. Since putting a sampled arm back in the sampling distribution only decreases the probability of the next sample being from a non-sampled cluster, the result we will derive will imply a lower bound for our problem.

Consider the following two-step sampling process, repeated for $j = 1, 2, 3, \ldots$, for the collection of arms: In the first step, a cluster $c \sim \boldsymbol{p}$ is drawn and then the arm $a_j$ with $c(a_j) = c$ is considered.

In the second step, the arm $a_j$ is kept in the set of collected arms with probability $\frac{p_{\min}}{p_c}$, and it is discarded with probability $1 - \frac{p_{\min}}{p_c}$. Let $l_j$ denote the outcome of the $j^{\text{th}}$ draw, so that $l_j = 0$ if the sampled arm was discarded and $l_j = c(a_j)$ if the arm was kept. Under this two-step sampling process, we have for $c \in [C]$ that

$$\Pr[l_j = c] = p_c \frac{p_{\min}}{p_c} = p_{\min}$$

and so, the corresponding sampling distribution for a draw is

$$\begin{aligned}
\boldsymbol{p}_0 &:= \big( \Pr[l_j = 0], \Pr[l_j = 1], \ldots, \Pr[l_j = C] \big) \\
&= \big( 1 - C p_{\min}, p_{\min}, \ldots, p_{\min} \big)
\end{aligned}$$

Thus, with regard to $[C]$, $\boldsymbol{p}_0$ is a uniform distribution. Then, the probability of the event that a new draw gives a new cluster when $i-1$ clusters have already been collected is $\frac{C-i+1}{C}(1-(1-Cp_{\min})) = (C - i + 1)p_{\min}$. Let $L$ denote the number of draws needed to cover all of the clusters. From properties of the geometric distribution it follows that

$$\begin{aligned}
\mathbb{E}[L] &= p_{\min}^{-1} \sum_{c \in [C]} \frac{1}{C - c + 1} \\
&= O(p_{\min}^{-1} \log C)
\end{aligned}$$

and

$$\begin{aligned}
Var[L] &< p_{\min}^{-2} \sum_{c \in [C]} \frac{1}{(C - c + 1)^2} \\
&< 2 p_{\min}^{-2}
\end{aligned}$$

By Cantelli's inequality we get

$$\begin{aligned}
\Pr[L \leqslant \mathbb{E}[L] + p_{\min}^{-1} T^{\delta}] &\geqslant 1 - \frac{Var[L]}{Var[L] + p_{\min}^{-2} T^{2\delta}} \\
&> 1 - \frac{1}{1 + \frac{1}{2} T^{2\delta}}
\end{aligned}$$

$\square$

The proof of the above lemma relies on arguments similar to those that can be used for the coupon collector problem (Flajolet et al., 1992). Once the set $\mathcal{S}$ is determined, in order to collect observations to perform clustering, each arm in $\mathcal{S}$ is played $T_0$ times, with the precise value of $T_0$ being defined later in this subsection. Even though a consumption vector is also observed every time an arm is played, for the purposes of the clustering only, we will utilize just the context and the reward, but we could as well have chosen to utilize one of the $d$ resources instead of the reward. In any case, the resource consumption due to these initial $N_{\mathcal{S}} \cdot T_0$ plays will still have to be subtracted from the budget.

The arms in $\mathcal{S}$ are clustered using the classifier-Lasso method proposed by (Su et al., 2016), which relies on the following objective function

$$\begin{aligned}
&Q\big((\boldsymbol{\mu}_a)_{a \in \mathcal{S}}, (\boldsymbol{\mu}_c)_{c \in [C]}\big) \\
&= \frac{1}{N_{\mathcal{S}} \cdot T_0} \sum_{a \in \mathcal{S}} \sum_{t : a_t = a} \frac{1}{2} \big(r_t(a) - \boldsymbol{\mu}_a^{\top} \boldsymbol{x}_t(a)\big)^2 \\
&\quad + \frac{\lambda_1}{N_{\mathcal{S}}} \sum_{a \in \mathcal{S}} \prod_{c \in [C]} \|\boldsymbol{\mu}_a - \boldsymbol{\mu}_c\|
\end{aligned} \tag{9}$$

where $\lambda_1 \in \mathbb{R}_+$ is a regularization parameter. It is valuable to point out that $N_{\mathcal{S}} + C$ vectors of parameters are estimated in total, since the method does not impose the parameter of an arm to be

corresponding to one of the cluster parameters. The estimation of the reward parameters follows by minimizing this objective function,

$$
\begin{aligned}
\left((\widehat{\boldsymbol{\mu}}_a)_{a\in\mathcal{S}}, (\widehat{\boldsymbol{\mu}}_c)_{c\in[C]}\right) \\
:= \underset{\left((\boldsymbol{\mu}_a)_{a\in\mathcal{S}}, (\boldsymbol{\mu}_c)_{c\in[C]}\right)}{\arg\min} Q\left((\boldsymbol{\mu}_a)_{a\in\mathcal{S}}, (\boldsymbol{\mu}_c)_{c\in[C]}\right).
\end{aligned}
\tag{10}
$$

Then, the arm $a \in S$ is clustered as

$$
\widehat{c}(a) := \sum_{c\in[C]} c \cdot \mathbb{1}[\widehat{\boldsymbol{\mu}}_a = \widehat{\boldsymbol{\mu}}_c]
\tag{11}
$$

Since under the classifier-Lasso method it is possible for an arm's estimated parameter to not be equal to the estimated parameter of any cluster, an arm can be assigned to none of the clusters, leading to the case where $\widehat{c}(a) = 0$. However, as shown in the proof of the focal result of this subsection, only few of the arms are not assigned to any cluster.

Besides the coverage of the clusters, we also want to achieve high clustering accuracy. We define the clustering error for cluster $c \in [C]$ as

$$
\epsilon_c := \frac{\sum_{a\in\mathcal{S}} \mathbb{1}[\widehat{c}(a) = c, c(a) \neq c]}{\sum_{a\in\mathcal{S}} \mathbb{1}[\widehat{c}(a) = c]}
\tag{12}
$$

Thus, the clustering error for $c$ is the proportion of arms assigned to $c$ that should have been assigned to other clusters. We want to ensure that for each cluster $c$, the output of Eq. (11) results to low error $\epsilon_c$. Towards this end, we make the following assumptions.

**Assumption A.5** (Separation). *We assume $\|\boldsymbol{\mu}_c - \boldsymbol{\mu}_{c'}\| \geq \xi_1 > 0$, for $c, c' \in [C], c \neq c'$.*

**Assumption A.6.** *We assume the number of clusters $C$ is fixed.*

**Assumption A.7.** *We assume*

$$
T_0 \lambda_1^2 / (\log T_0)^{6+2\xi_2} \to \infty,
$$

*and*

$$
\lambda_1 (\log T_0)^{\xi_2} \to 0,
$$

*for $\xi_2 > 0$ as $(N_\mathcal{S}, T_0) \to \infty$.*

**Assumption A.8.** *We assume*

$$
N_\mathcal{S}^{1/2} T_0^{-1} (\log T_0)^9 \to 0,
$$

*and*

$$
N_\mathcal{S}^2 T_0^{1-\xi_3/2} \to \xi_4 < \infty,
$$

*for $\xi_3 \geq 6$ as $(N_\mathcal{S}, T_0) \to \infty$.*

Assumption A.5 is needed to ensure that the clusters are distinguishable, Assumption A.6 disallows the number of clusters to grow asymptotically. While Assumptions A.7 and A.8 impose appropriate rates on $\lambda_1, N_\mathcal{S}$, and $T_0$, the algorithm need to perform clustering based on one of the $d$ resources instead of the reward.

**Lemma A.9.** *Under Assumptions 3.1-3.4, A.5-A.8, the clustering error is $\epsilon_c = o(p_{\min}^{-1} N_\mathcal{S}^{-1})$ with high probability, for any $c \in [C]$.*

*Proof.* In this proof, we will exploit the Azuma-Hoeffding inequality and a result from (Su et al., 2016) that holds under the stated assumptions.

Now, for the clustering error $\epsilon_c$ we have

$$
\begin{aligned}
\epsilon_c &= \frac{\sum_{a\in\mathcal{S}} \mathbb{1}[\widehat{c}(a) = c, c(a) \neq c]}{\sum_{a\in\mathcal{S}} \mathbb{1}[\widehat{c}(a) = c]} \\
&= \frac{\sum_{a\in\mathcal{S}} \mathbb{1}[\widehat{c}(a) = c, c(a) \neq c]}{\sum_{a\in\mathcal{S}} \mathbb{1}[c(a) = c] - \sum_{a\in\mathcal{S}} \mathbb{1}[\widehat{c}(a) \neq c, c(a) = c] + \sum_{a\in\mathcal{S}} \mathbb{1}[\widehat{c}(a) = c, c(a) \neq c]}
\end{aligned}
$$

$$\leqslant \frac{\sum_{a\in\mathcal{S}} \mathbb{1}[\widehat{c}(a)=c, c(a)\neq c]}{\sum_{a\in\mathcal{S}} \mathbb{1}[c(a)=c] - \sum_{a\in\mathcal{S}} \mathbb{1}[\widehat{c}(a)\neq c, c(a)=c]}$$

$$= \frac{\sum_{a\in\mathcal{S}} \mathbb{1}[\widehat{c}(a)=c, c(a)\neq c]}{\sum_{a:c(a)=c} 1 - \mathbb{1}[\widehat{c}(a)\neq c]}$$

$$\leqslant \frac{\sum_{a\in\mathcal{S}} \mathbb{1}[\widehat{c}(a)=c, c(a)\neq c]}{O(1)\,\mathbb{E}\left[\sum_{a:c(a)=c} 1 - \mathbb{1}[\widehat{c}(a)\neq c]\right]} \tag{13}$$

$$= \frac{\sum_{a\in\mathcal{S}} \mathbb{1}[\widehat{c}(a)=c, c(a)\neq c]}{O(1)\,\mathbb{E}\left[\sum_{a\in\mathcal{S}} \mathbb{1}[c(a)=c, \widehat{c}(a)=c]\right]}$$

$$= \frac{\sum_{a\in\mathcal{S}} \mathbb{1}[\widehat{c}(a)=c, c(a)\neq c]}{O(1)\sum_{a\in\mathcal{S}} \Pr[c(a)=c]\Pr[\widehat{c}(a)=c|c(a)=c]}$$

$$= \frac{\sum_{a\in\mathcal{S}} \mathbb{1}[\widehat{c}(a)=c, c(a)\neq c]}{O(1)N_{\mathcal{S}}p_c(1-o(N_{\mathcal{S}}^{-1}))}$$

$$\leqslant \frac{\sum_{a\in\mathcal{S}} \mathbb{1}[\widehat{c}(a)=c, c(a)\neq c]}{O(1)N_{\mathcal{S}}p_{\min}(1-o(N_{\mathcal{S}}^{-1}))}$$

$$\leqslant O(1)\frac{\mathbb{E}\left[\sum_{a\in\mathcal{S}} \mathbb{1}[\widehat{c}(a)=c, c(a)\neq c]\right]}{N_{\mathcal{S}}p_{\min}(1-o(N_{\mathcal{S}}^{-1}))} \tag{14}$$

$$= O(1)\frac{\sum_{a\in\mathcal{S}} \Pr[c(a)\neq c]\Pr[\widehat{c}(a)=c|c(a)\neq c]}{N_{\mathcal{S}}p_{\min}(1-o(N_{\mathcal{S}}^{-1}))}$$

$$\leqslant O(1)\frac{\sum_{a\in\mathcal{S}} \Pr[c(a)\neq c]\Pr[\widehat{c}(a)\neq c(a)]}{N_{\mathcal{S}}p_{\min}(1-o(N_{\mathcal{S}}^{-1}))}$$

$$= O(1)\frac{N_{\mathcal{S}}(1-p_c)o(N_{\mathcal{S}}^{-1})}{N_{\mathcal{S}}p_{\min}(1-o(N_{\mathcal{S}}^{-1}))}$$

$$= O(1)\frac{(1-p_c)o(N_{\mathcal{S}}^{-1})}{p_{\min}(1-o(N_{\mathcal{S}}^{-1}))}$$

$$\leqslant \frac{o(p_{\min}^{-1}N_{\mathcal{S}}^{-1})}{1-o(N_{\mathcal{S}}^{-1})}$$

$$\leqslant o(p_{\min}^{-1}N_{\mathcal{S}}^{-1})$$

where Eq. (13) and (14) follow from the Azuma-Hoeffding inequality. □

In the proof of Lemma A.9 we exploit the fact that $\Pr[\widehat{c}(a)=c(a)] = 1 - o(N_{\mathcal{S}}^{-1})$ (Su et al., 2016, pp. 2250). Since the clustering error is, in asymptotic terms, the same for all clusters, we shall use $\epsilon_c$ to refer to this error for any cluster. Moreover, since $N_{\mathcal{S}}$ is given by Lemma A.4, the number of samples required from each arm in $\mathcal{S}$ for the clustering can be specified by satisfying Assumption A.8.

**Claim A.10.** *For $T_0 = N_{\mathcal{S}}$, Assumption A.8 is satisfied.*

A.3 ONLINE LEARNING

A special case of the online convex optimization problem is the game where at period $t \in [T]$ a learner chooses

$$\boldsymbol{\theta}_t \in \{\boldsymbol{\theta} \in [0,1]^d : \|\boldsymbol{\theta}\|_1 \leqslant 1\}$$

based on past observations, and the adversary chooses the learner's payoff to be the outcome of a function that is linear in $\boldsymbol{\theta}_t$. It is known (Srebro et al., 2011; Shalev-Shwartz et al., 2011) that for $\boldsymbol{\theta}_t$ chosen based on the online mirror descent algorithm, the regret against the best fixed action of the learner in the online convex optimization problem is $O(\sqrt{\log(d)T})$. In the context of our problem, we utilize this result following the approach of (Agrawal & Devanur, 2016). In particular, $\boldsymbol{\theta}_t$ will allow the consideration of resources in the arm choices in periods $t > N_{\mathcal{S}}T_0$, decreasing thus the probability that a choice will lead to depletion of one of the $d$ resources and consequently to the

termination of the algorithm. To derive the regret for our problem (Eq. (4)), we will consider as the payoff chosen by the adversary the value

$$\boldsymbol{\theta}_t^\top \left( \boldsymbol{v}_t(a_t) - \frac{B - N_\mathcal{S} T_0}{T - N_\mathcal{S} T_0} \mathbf{1}_d \right),$$

as illustrated by the algorithm presented later.

### A.4 CONFIDENCE ELLIPSOID

In this subsection we present results about bounds on the parameter estimates that will be derived by the algorithm after the clustering is conducted. The approach here is to express the clustering error as violation of the zero conditional mean assumption about the error terms. Then, the technique of confidence ellipsoids, which is standard in the literature of online learning, is employed to bound the parameter estimates. The definition that follows considers a linear model for a response variable $y_t$ that generalizes the response variables in the problem we consider, i.e., $r_t(a)$ and each dimension of $\boldsymbol{v}_t(a)$. Therefore, the results derived in this subsection will imply results about our actual problem.

This definition corresponds to a measurement error model (Wansbeek & Meijer, 2001; Fuller, 2009), as the expected value of the error term is $\mathbb{E}[\eta_t | \boldsymbol{x}_t] = \epsilon \boldsymbol{\gamma}^\top \boldsymbol{x}_t$, which in the general case is not zero. The error term $u_t$ corresponds to an error term $u_t$ of the model of our actual problem, i.e., $u_t$ represents $g_t(a)$ and the individual dimensions of $\boldsymbol{q}_t(a)$. Since the context is i.i.d. across arms and periods, the probability $\epsilon$ is perceived as the clustering error in Eq. (12). Thus, $\boldsymbol{\gamma}$ is perceived as the element-wise difference between parameters of different clusters. For instance, considering $C = 2$ and $y_t$ corresponding to $r_t(a)$ for $c(a) = 1$, we have that $\boldsymbol{\gamma}$ corresponds to $\boldsymbol{\mu}_2 - \boldsymbol{\mu}_1$. Therefore, $h_t$ represents the part of the error term due to $\boldsymbol{x}_t$ being endogenous, when zero clustering error is falsely assumed. We let $h_t$ have two branches instead of $C$ because $\boldsymbol{\gamma}$ can be perceived as a bound on parameter differences.

This confidence ellipsoid captures the set of plausible values for the unknown parameter vector $\boldsymbol{\mu}$, given the observations up to time $t$. The radius $\rho_t$ consists of three terms: a statistical concentration term accounting for zero-mean stochastic noise $u_t$ (as in Assumption 3.2), a bias term $\epsilon m \sqrt{t}$ introduced by clustering errors, and a regularization term $\sqrt{\lambda_2 m}$ reflecting uncertainty due to the use of ridge regression. With high probability (at least $1 - \zeta$), the true parameter vector $\boldsymbol{\mu}$ lies within $\mathcal{C}_t$.

The next two lemmas are the results derived in this subsection.

**Lemma A.11.** *Under the zero-mean and bounded-noise assumptions (Assumption 3.2 and 3.3) and with regularization parameter $\lambda_2$, define the confidence radiue*

$$\rho_t := 2(R+1)\sqrt{m \log\left(\frac{tm}{\lambda_2 \zeta}\right)} + \epsilon m \sqrt{t} + \sqrt{\lambda_2 m}.$$

*Then, for any time $t \in [T]$ with probability at least $1 - \zeta$, the true parameter vector $\boldsymbol{\mu}$ satisfies*

$$\|\mu - \widehat{\mu}_t\|_{\boldsymbol{M}_t} \leqslant \rho_t$$

*Proof.* By lemmas B.4, B.5, B.6, and B.7, we have that for any $\zeta \in (0, 1)$, with probability at least $1 - \zeta$,

$$|\boldsymbol{x}^\top \widehat{\boldsymbol{\mu}}_t - \boldsymbol{x}^\top \boldsymbol{\mu}| \leqslant \|\boldsymbol{x}\|_{\boldsymbol{M}_t^{-1}} \cdot A_1$$

where

$$A_1 := \Big( (R+1)\sqrt{2 \log\big(\det(\boldsymbol{M}_t)^{1/2} \det(\lambda_2 \boldsymbol{I}_m)^{-1/2}/\zeta\big)} + \epsilon \bar{\gamma} m \sqrt{t} + \sqrt{\lambda_2 m} \Big)$$

Now, by letting $\boldsymbol{x} = \boldsymbol{M}_t(\widehat{\boldsymbol{\mu}}_t - \boldsymbol{\mu})$, we have

$$\|\widehat{\boldsymbol{\mu}}_t - \boldsymbol{\mu}\|_{\boldsymbol{M}_t}^2 \leqslant \|\boldsymbol{M}_t(\widehat{\boldsymbol{\mu}}_t - \boldsymbol{\mu})\|_{\boldsymbol{M}_t^{-1}} \cdot A_1$$

$$= \|\widehat{\boldsymbol{\mu}}_t - \boldsymbol{\mu}\|_{\boldsymbol{M}_t} \cdot A_1$$

Dividing both sides by $\|\widehat{\boldsymbol{\mu}}_t - \boldsymbol{\mu}\|_{\boldsymbol{M}_t}$ we get

$$\|\widehat{\boldsymbol{\mu}}_t - \boldsymbol{\mu}\|_{\boldsymbol{M}_t} \leqslant A_1 \tag{15}$$

Since $\|\boldsymbol{x}_t\|_2 \leqslant \sqrt{m}$, and $\boldsymbol{M}_t$ and $\lambda_2 \boldsymbol{I}_m$ are positive-definite matrices, we can upper bound the first term in $A_1$ (ignoring the term $R + 1$) as follows

$$\sqrt{2\log\Big(\frac{\det(\boldsymbol{M}_t)^{1/2}}{\det(\lambda_2 \boldsymbol{I}_m)^{1/2}/\zeta}\Big)} \leqslant \sqrt{m\log\Big(\frac{1 + tm/\lambda_2}{\zeta}\Big)} \tag{16}$$

and thus

$$\|\widehat{\boldsymbol{\mu}}_t - \boldsymbol{\mu}\|_{\boldsymbol{M}_t}$$

$$\leqslant (R + 1)\sqrt{m\log\Big(\frac{1 + tm/\lambda_2}{\zeta}\Big)} + \epsilon\bar{\gamma}m\sqrt{t} + \sqrt{\lambda_2 m} \tag{17}$$

$$\leqslant 2(R + 1)\sqrt{m\log\Big(\frac{tm}{\lambda_2\zeta}\Big)} + \epsilon m\sqrt{t} + \sqrt{\lambda_2 m} \tag{18}$$

where $\bar{\gamma} \leqslant 1$. $\qquad\qquad\square$

**Lemma A.12** (Sum of rewards). *Consider $\widetilde{\boldsymbol{\mu}}_t \in \mathcal{C}_t$. For $R = \frac{1}{2}$ and $\lambda_2 = 1$, with probability at least $1 - \zeta$*

$$\sum_{t=1}^{T} |\widetilde{\boldsymbol{\mu}}_t^\top \boldsymbol{x}_t - \boldsymbol{\mu}^\top \boldsymbol{x}_t|$$

$$\leqslant 4m\sqrt{T\log(Tm/\zeta)\log(T)} + \epsilon m^{\frac{3}{2}}T\sqrt{\log(T)}.$$

Lemma A.11 implies that a parameter estimate lies within some fixed distance from the true parameter with some specified probability, while Lemma A.12 serves as the basis for employing optimistic parameter estimates, as described later.

# B USEFUL TOOLS FOR SECTION A

Here in this section, we provide some missing proofs for Section A. In Section B.1 we provide some lemmas for Lemma A.11. In Section B.2 we provide proof for Lemma A.12.

## B.1 PROOF OF LEMMA A.11

For notational convenience, let

**Definition B.1.**

$$\boldsymbol{X} := (\boldsymbol{x}_1^\top, \ldots, \boldsymbol{x}_t^\top) \in [0,1]^{t \times m}$$

$$\boldsymbol{y} := (y_1, \ldots, y_t) \in [0,1]^t$$

$$\boldsymbol{\eta} := (\eta_1, \ldots, \eta_t) \in \mathbb{R}^t$$

$$\boldsymbol{s}_t := \sum_{i=1}^{t-1} \boldsymbol{x}_i(\eta_i - \mathbb{E}[\eta_i|\boldsymbol{x}_i])$$

$$\bar{\gamma} := \|\boldsymbol{\gamma}\|_\infty$$

$$\langle \boldsymbol{a}, \boldsymbol{b} \rangle_{\boldsymbol{M}} := \boldsymbol{a}^\top \boldsymbol{M} \boldsymbol{b}, \text{ for } \boldsymbol{a}, \boldsymbol{b} \in \mathbb{R}^m, \boldsymbol{M} \in \mathbb{R}^{m \times m}$$

For the proof we will need the following two Facts and four Lemmas.

**Fact B.2.** *We can show that* $\widehat{\boldsymbol{\mu}}_t = \boldsymbol{\mu} - \lambda_2 \boldsymbol{M}_t^{-1} \boldsymbol{\mu} + \boldsymbol{M}_t^{-1} \boldsymbol{X}^\top \boldsymbol{\eta}$

*Proof.*

$$
\begin{aligned}
\widehat{\boldsymbol{\mu}}_t &= \boldsymbol{M}_t^{-1} \boldsymbol{X}^\top \boldsymbol{y} \\
&= \boldsymbol{M}_t^{-1} \boldsymbol{X}^\top (\boldsymbol{X}\boldsymbol{\mu} + \boldsymbol{\eta}) \\
&= \boldsymbol{M}_t^{-1} \boldsymbol{X}^\top \boldsymbol{X}\boldsymbol{\mu} + \boldsymbol{M}_t^{-1} \boldsymbol{X}^\top \boldsymbol{\eta} \\
&= \boldsymbol{M}_t^{-1} \boldsymbol{X}^\top \boldsymbol{X}\boldsymbol{\mu} + \boldsymbol{M}_t^{-1} \lambda_2 \boldsymbol{I}_m \boldsymbol{\mu} - \boldsymbol{M}_t^{-1} \lambda_2 \boldsymbol{I}_m \boldsymbol{\mu} + \boldsymbol{M}_t^{-1} \boldsymbol{X}^\top \boldsymbol{\eta} \\
&= \boldsymbol{M}_t^{-1} \boldsymbol{M}_t \boldsymbol{\mu} - \lambda_2 \boldsymbol{M}_t^{-1} \boldsymbol{\mu} + \boldsymbol{M}_t^{-1} \boldsymbol{X}^\top \boldsymbol{\eta} \\
&= \boldsymbol{\mu} - \lambda_2 \boldsymbol{M}_t^{-1} \boldsymbol{\mu} + \boldsymbol{M}_t^{-1} \boldsymbol{X}^\top \boldsymbol{\eta}
\end{aligned}
$$

Thus, we complete the proof. $\qquad\square$

**Fact B.3.** *For any* $\boldsymbol{x} \in \mathbb{R}^m$, *we have that* $\boldsymbol{x}^\top \widehat{\boldsymbol{\mu}}_t - \boldsymbol{x}^\top \boldsymbol{\mu} = \langle \boldsymbol{x}, \boldsymbol{X}^\top \boldsymbol{\eta} \rangle_{\boldsymbol{M}_t^{-1}} - \lambda_2 \langle \boldsymbol{x}, \boldsymbol{\mu} \rangle_{\boldsymbol{M}_t^{-1}}$

*Proof.* From Fact B.2 we have

$$
\begin{aligned}
\boldsymbol{x}^\top \widehat{\boldsymbol{\mu}}_t - \boldsymbol{x}^\top \boldsymbol{\mu} &= \boldsymbol{x}^\top \left( \boldsymbol{\mu} - \lambda_2 \boldsymbol{M}_t^{-1} \boldsymbol{\mu} + \boldsymbol{M}_t^{-1} \boldsymbol{X}^\top \boldsymbol{\eta} \right) - \boldsymbol{x}^\top \boldsymbol{\mu} \\
&= \boldsymbol{x}^\top \boldsymbol{M}_t^{-1} \boldsymbol{X}^\top \boldsymbol{\eta} - \lambda_2 \boldsymbol{x}^\top \boldsymbol{M}_t^{-1} \boldsymbol{\mu} \\
&= \langle \boldsymbol{x}, \boldsymbol{X}^\top \boldsymbol{\eta} \rangle_{\boldsymbol{M}_t^{-1}} - \lambda_2 \langle \boldsymbol{x}, \boldsymbol{\mu} \rangle_{\boldsymbol{M}_t^{-1}}
\end{aligned}
$$

Thus, we complete the proof. $\qquad\square$

**Lemma B.4.** *We can show that*

$$
|\boldsymbol{x}^\top \widehat{\boldsymbol{\mu}}_t - \boldsymbol{x}^\top \boldsymbol{\mu}| \leqslant \|\boldsymbol{x}\|_{\boldsymbol{M}_t^{-1}} \left( \|\boldsymbol{s_t}\|_{\boldsymbol{M}_t^{-1}} + \epsilon \|\boldsymbol{X}^\top \boldsymbol{X} \boldsymbol{\gamma}\|_{\boldsymbol{M}_t^{-1}} + \sqrt{\lambda_2 m} \right)
$$

*Proof.* Using the Cauchy-Schwarz inequality on Fact B.3, we have

$$
|\boldsymbol{x}^\top \widehat{\boldsymbol{\mu}}_t - \boldsymbol{x}^\top \boldsymbol{\mu}| \leqslant \|\boldsymbol{x}\|_{\boldsymbol{M}_t^{-1}} \left( \|\boldsymbol{X}^\top \boldsymbol{\eta}\|_{\boldsymbol{M}_t^{-1}} + \lambda_2 \|\boldsymbol{\mu}\|_{\boldsymbol{M}_t^{-1}} \right)
$$

We can upper bound the second term of the above equation as follows:

$$
\begin{aligned}
\|\boldsymbol{X}^\top \boldsymbol{\eta}\|_{\boldsymbol{M}_t^{-1}} + \lambda_2 \|\boldsymbol{\mu}\|_{\boldsymbol{M}_t^{-1}} &\leqslant \|\boldsymbol{X}^\top \boldsymbol{\eta}\|_{\boldsymbol{M}_t^{-1}} + \sqrt{\lambda_2} \|\boldsymbol{\mu}\|_2 \\
&\leqslant \|\boldsymbol{X}^\top \boldsymbol{\eta}\|_{\boldsymbol{M}_t^{-1}} + \sqrt{\lambda_2 m} \\
&= \|\boldsymbol{X}^\top (\boldsymbol{\eta} + \mathbb{E}[\boldsymbol{\eta}|\boldsymbol{X}] - \mathbb{E}[\boldsymbol{\eta}|\boldsymbol{X}])\|_{\boldsymbol{M}_t^{-1}} + \sqrt{\lambda_2 m} \\
&= \|\boldsymbol{s_t} + \boldsymbol{X}^\top \mathbb{E}[\boldsymbol{\eta}|\boldsymbol{X}]\|_{\boldsymbol{M}_t^{-1}} + \sqrt{\lambda_2 m} \\
&= \|\boldsymbol{s_t} + \epsilon \boldsymbol{X}^\top \boldsymbol{X} \boldsymbol{\gamma}\|_{\boldsymbol{M}_t^{-1}} + \sqrt{\lambda_2 m} \\
&\leqslant \|\boldsymbol{s_t}\|_{\boldsymbol{M}_t^{-1}} + \|\epsilon \boldsymbol{X}^\top \boldsymbol{X} \boldsymbol{\gamma}\|_{\boldsymbol{M}_t^{-1}} + \sqrt{\lambda_2 m} \\
&= \|\boldsymbol{s_t}\|_{\boldsymbol{M}_t^{-1}} + \epsilon \|\boldsymbol{X}^\top \boldsymbol{X} \boldsymbol{\gamma}\|_{\boldsymbol{M}_t^{-1}} + \sqrt{\lambda_2 m}
\end{aligned}
$$

where the second inequality follows from that $\|\boldsymbol{\mu}\|_{\boldsymbol{M}_t^{-1}}^2 \leqslant \frac{1}{\lambda_{\min}(\boldsymbol{M}_t) \|\boldsymbol{\mu}\|_2^2} \leqslant \frac{1}{\lambda_2 \|\boldsymbol{\mu}\|_2^2}$. $\qquad\square$

**Lemma B.5.** *Consider the* $\sigma-$*algebra* $F_t = \sigma(\boldsymbol{x}_1, \ldots, \boldsymbol{x}_t, \eta_1, \ldots, \eta_{t-1})$, *such that* $\{F_t\}_{t=1}^\infty$ *is a filtration, and* $\eta_t - \mathbb{E}[\eta_t|\boldsymbol{x}_t]$ *is* $F_t$-*measurable. Then,* $\eta_t - \mathbb{E}[\eta_t|\boldsymbol{x}_t]$ *is* $(R+1)$-*sub-Gaussian.*

*Proof.* It suffices to show that $\eta_t - \mathbb{E}[\eta_t | \boldsymbol{x}_t]$ lies in an interval of length at most $2(R+1)$. We can upper bound $|\eta_t - \mathbb{E}[\eta_t | \boldsymbol{x}_t]|$ as follows:

$$
\begin{aligned}
|\eta_t - \mathbb{E}[\eta_t | \boldsymbol{x}_t]| &= |u_t + h_t - \epsilon \boldsymbol{\gamma}^\top \boldsymbol{x}_t| \\
&\leqslant |u_t| + |h_t| + |\epsilon \boldsymbol{\gamma}^\top \boldsymbol{x}_t| \\
&\leqslant 2R + (1 + \epsilon) |\boldsymbol{\gamma}^\top \boldsymbol{x}_t| \\
&\leqslant 2(R+1)
\end{aligned}
$$

$\square$

**Lemma B.6.** *For any $\zeta \in (0, 1)$, with probability at least $1 - \zeta$, if $\eta_t - \mathbb{E}[\eta_t | \boldsymbol{x}_t]$ is $(R+1)$-sub-Gaussian, then for all $t > 0$,*

$$
\|\boldsymbol{s}_t\|_{\boldsymbol{M}_t^{-1}}^2 \leqslant 2(R+1)^2 \log \left( \det(\boldsymbol{M}_t)^{1/2} \det(\lambda_2 \boldsymbol{I}_m)^{-1/2} / \zeta \right)
$$

*Proof.* We apply Theorem 1 of (Abbasi-Yadkori et al., 2011) with $\eta_t - \mathbb{E}[\eta_t | \boldsymbol{x}_t]$ in the place of $\eta_t$. $\square$

**Lemma B.7.** *We can show that*

$$
\|\boldsymbol{X}^\top \boldsymbol{X} \boldsymbol{\gamma}\|_{\boldsymbol{M}_t^{-1}} \leqslant \bar{\gamma} m \sqrt{t}
$$

*Proof.* Since $\boldsymbol{X}^\top \boldsymbol{X} \in \mathbb{R}^{m \times m}$ contains non-negative entries, we have

$$
\|\boldsymbol{X}^\top \boldsymbol{X} \boldsymbol{\gamma}\|_{\boldsymbol{M}_t^{-1}} \leqslant \bar{\gamma} \cdot \|\boldsymbol{X}^\top \boldsymbol{X} \boldsymbol{1}_m\|_{\boldsymbol{M}_t^{-1}}
$$

Next, we just need to upper bound $\|\boldsymbol{X}^\top \boldsymbol{X} \boldsymbol{1}_m\|_{\boldsymbol{M}_t^{-1}}^2$. By the properties of PSD/PD matrices, we know that

$$
\lambda_2 \boldsymbol{I}_m + \boldsymbol{X}^\top \boldsymbol{X} > \boldsymbol{X}^\top \boldsymbol{X} \succeq 0
$$

which implies that

$$
(\boldsymbol{X}^\top \boldsymbol{X})^{1/2} \cdot (\lambda_2 \boldsymbol{I}_m + \boldsymbol{X}^\top \boldsymbol{X})^{-1} \cdot (\boldsymbol{X}^\top \boldsymbol{X})^{1/2} < \boldsymbol{I}_m
$$

Thus, we have

$$
\begin{aligned}
\boldsymbol{1}_m^\top \boldsymbol{X}^\top \boldsymbol{X} (\lambda_2 \boldsymbol{I}_m + \boldsymbol{X}^\top \boldsymbol{X})^{-1} \boldsymbol{X}^\top \boldsymbol{X} \boldsymbol{1}_m &\leqslant \boldsymbol{1}_m^\top \boldsymbol{X}^\top \boldsymbol{X} \boldsymbol{1}_m \\
&= \|\boldsymbol{X} \boldsymbol{1}_m\|_2^2 \\
&\leqslant t m^2
\end{aligned}
$$

where the last step follows since each entry of $\boldsymbol{X}$ is between $0$ and $1$. $\square$

## B.2 PROOF OF LEMMA A.12

We will need the following fact and lemma.

**Fact B.8.** *For any positive definite matrix $\boldsymbol{M} \in \mathbb{R}^{m \times m}$ and any two vectors $\boldsymbol{a}, \boldsymbol{b} \in \mathbb{R}^m$, it holds that $|\boldsymbol{a}^\top \boldsymbol{b}| \leqslant \|\boldsymbol{a}\|_{\boldsymbol{M}} \|\boldsymbol{b}\|_{\boldsymbol{M}^{-1}}$.*

**Lemma B.9** (Lemma 3 of (Agrawal & Devanur, 2016)). *For $\boldsymbol{x}_i \in \mathbb{R}^m$ with $\|\boldsymbol{x}_i\|_2 \leqslant \sqrt{m}$, it holds that*

$$
\sum_{i=1}^t \|\boldsymbol{x}_i\|_{\boldsymbol{M}_i^{-1}} \leqslant \sqrt{mt \log(t)}
$$

Now, we derive the statement of the lemma as:

$$
\sum_{t=1}^T |\tilde{\boldsymbol{\mu}}_t^\top \boldsymbol{x}_t - \boldsymbol{\mu}^\top \boldsymbol{x}_t| \leqslant \sum_{t=1}^T \|\boldsymbol{x}_t\|_{\boldsymbol{M}_t^{-1}} \|\tilde{\boldsymbol{\mu}}_t - \boldsymbol{\mu}\|_{\boldsymbol{M}_t}
$$

$$\leqslant \left(3\sqrt{m\log\left(Tm/\zeta\right)} + \epsilon m\sqrt{T} + \sqrt{m}\right)\sum_{t=1}^{T}\|\boldsymbol{x}_t\|_{\boldsymbol{M}_t^{-1}}$$

$$\leqslant \left(4\sqrt{m\log\left(Tm/\zeta\right)} + \epsilon m\sqrt{T}\right)\sum_{t=1}^{T}\|\boldsymbol{x}_t\|_{\boldsymbol{M}_t^{-1}}$$

$$\leqslant \left(4\sqrt{m\log\left(Tm/\zeta\right)} + \epsilon m\sqrt{T}\right)\sqrt{mT\log(T)}$$

$$= 4m\sqrt{T\log\left(Tm/\zeta\right)\log(T)} + \epsilon m^{\frac{3}{2}}T\sqrt{\log(T)}$$

where the first step follows from Fact B.8, the second from Lemma A.11, and the fourth from Lemma B.9.

## C  USEFUL LEMMAS FOR SECTION 4

Here we provide the following lemmas.

**Lemma C.1** (Informal version of Lemma 4.4). *Given clustering* $\{\widehat{c}(a)\}_{a\in\mathcal{S}}$ *and vectors* $\{\boldsymbol{\theta}_i\}_{i=N_{\mathcal{S}}\cdot T_0+1}^{t}$, *where* $\boldsymbol{\theta}_i \in [0,1]^d$, *with probability at least* $1-\zeta$ *we have that for any* $a \in \mathcal{S}$,

a) $\boldsymbol{x}_t(a)^\top\left(\widetilde{\boldsymbol{\mu}}_{a,t} - \boldsymbol{\mu}_{\widehat{c}(a)}\right) \geqslant 0$

b) $\boldsymbol{x}_t(a)^\top\left(\widetilde{\boldsymbol{W}}_{a,t} - \boldsymbol{W}_{\widehat{c}(a)}\right)\boldsymbol{\theta}_t \leqslant 0$

c) $|\sum_{i=N_{\mathcal{S}}\cdot T_0+1}^{t}\boldsymbol{x}_i(a_i)^\top(\widetilde{\boldsymbol{\mu}}_{a_i,i} - \boldsymbol{\mu}_{\widehat{c}(a_i)})| \leqslant \rho$

d) $\|\sum_{i=N_{\mathcal{S}}\cdot T_0+1}^{t}\boldsymbol{x}_i(a_i)^\top(\widetilde{\boldsymbol{W}}_{a_i,i} - \boldsymbol{W}_{\widehat{c}(a_i)})\|_\infty \leqslant \rho$

*Where $\rho$ is given by:*

$$\rho := 4Cm\sqrt{t\log(tm/\zeta)\log(t)} + \epsilon_c m^{\frac{3}{2}}t\sqrt{\log(t)}$$

*Proof.* Statements $a)$ and $b)$ follow directly from Eq. (6) and (7). For statement $c)$ we have:

$$|\sum_{i=N_{\mathcal{S}}\cdot T_0+1}^{t}\boldsymbol{x}_i(a_i)^\top(\widetilde{\boldsymbol{\mu}}_{a_i,i} - \boldsymbol{\mu}_{\widehat{c}(a_i)})| \leqslant \sum_{i=N_{\mathcal{S}}\cdot T_0+1}^{t}|\boldsymbol{x}_i(a_i)^\top(\widetilde{\boldsymbol{\mu}}_{a_i,i} - \boldsymbol{\mu}_{\widehat{c}(a_i)})|$$

$$= \sum_{c\in[C]}\sum_{\substack{i:\widehat{c}(a_i)=c \\ i>N_{\mathcal{S}}\cdot T_0}}|\boldsymbol{x}_i(a_i)^\top(\widetilde{\boldsymbol{\mu}}_{a_i,i} - \boldsymbol{\mu}_{c})|$$

$$\leqslant \sum_{c\in[C]}4m\sqrt{t_c\log\left(t_cm/\zeta\right)\log(t_c)} + \epsilon_c m^{\frac{3}{2}}t_c\sqrt{\log(t_c)}$$

$$\leqslant 4Cm\sqrt{t\log\left(tm/\zeta\right)\log(t)} + \epsilon_c m^{\frac{3}{2}}t\sqrt{\log(t)}$$

where the first step follows from the triangle inequality, and the third from Lemma A.12. Statement $d)$ follows similarly. $\qquad\square$

## D  USEFUL LEMMAS FOR SECTION 5

Here we provide the following results. In Section D.1 we provide the lemmas for Lemma 5.5. In Section D.2 we provide the lemmas for Theorem D.8.

### D.1  USEFUL LEMMAS FOR LEMMA 5.5

**Lemma D.1** (Formal version of Lemma 5.2). *If* $\max_{\pi\in\{\pi*,\pi'\}}|r(\pi) - \rho(\pi)| \leqslant \epsilon'$ *then* $|r(\pi^*) - \rho(\pi')| \leqslant \epsilon'$.

*Proof.* Considering $\pi^*$ we have

$$r(\pi^*) - \rho(\pi^*) \leqslant \epsilon'$$

which implies that

$$r(\pi^*) - \rho(\pi') \leqslant \epsilon'$$

and by considering $\pi'$ we similarly get

$$\rho(\pi') - r(\pi^*) \leqslant \epsilon'$$

Thus, it follows that $|r(\pi^*) - \rho(\pi')| \leqslant \epsilon'$. $\qquad\square$

Therefore, it suffices to prove that the difference $\max_{\pi \in \{\pi^*, \pi'\}} |r(\pi) - \rho(\pi)|$ is small. We first consider $\pi^*$ and then $\pi'$.

**Lemma D.2** (Formal version of Lemma 5.3). $|r(\pi^*) - \rho(\pi^*)| < o(1)$

*Proof.*

$$|r(\pi^*) - \rho(\pi^*)|$$

$$\leqslant |\mathop{\mathbb{E}}_{\boldsymbol{X}} \Big[ \sum_{a \in [K]} \boldsymbol{\mu}_{c(a)}^\top \boldsymbol{x}(a) \pi^*(a, \boldsymbol{X}) \Big] - \frac{K}{N_\mathcal{S}} \mathop{\mathbb{E}}_{\boldsymbol{X}} \Big[ \sum_{a \in \mathcal{S}} \boldsymbol{\mu}_{c(a)}^\top \boldsymbol{x}(a) \pi^*(a, \boldsymbol{X}) \Big]|$$

$$+ |\frac{K}{N_\mathcal{S}} \mathop{\mathbb{E}}_{\boldsymbol{X}} \Big[ \sum_{a \in \mathcal{S}} \boldsymbol{\mu}_{c(a)}^\top \boldsymbol{x}(a) \pi^*(a, \boldsymbol{X}) \Big] - \frac{K}{N_\mathcal{S}} \mathop{\mathbb{E}}_{\boldsymbol{X}} \Big[ \sum_{a \in \mathcal{S}} \boldsymbol{\mu}_{\hat{c}(a)}^\top \boldsymbol{x}(a) \pi^*(a, \boldsymbol{X}) \Big]|$$

$$+ |\frac{K}{N_\mathcal{S}} \mathop{\mathbb{E}}_{\boldsymbol{X}} \Big[ \sum_{a \in \mathcal{S}} \boldsymbol{\mu}_{\hat{c}(a)}^\top \boldsymbol{x}(a) \pi^*(a, \boldsymbol{X}) \Big] - \frac{K}{N_\mathcal{S}} \mathop{\mathbb{E}}_{\boldsymbol{X}} \Big[ \sum_{a \in \mathcal{S}} \hat{\boldsymbol{\mu}}_{\hat{c}(a), N_\mathcal{S} T_0 + 1}^\top \boldsymbol{x}(a) \pi^*(a, \boldsymbol{X}) \Big]|$$

$$+ |\frac{K}{N_\mathcal{S}} \mathop{\mathbb{E}}_{\boldsymbol{X}} \Big[ \sum_{a \in \mathcal{S}} \hat{\boldsymbol{\mu}}_{\hat{c}(a), N_\mathcal{S} T_0 + 1}^\top \boldsymbol{x}(a) \pi^*(a, \boldsymbol{X}) \Big] - \frac{K}{N_\mathcal{S}} \frac{1}{N_\mathcal{S} T_0} \sum_{t=1}^{N_\mathcal{S} T_0} \sum_{a \in \mathcal{S}} \hat{\boldsymbol{\mu}}_{\hat{c}(a), N_\mathcal{S} T_0 + 1}^\top \boldsymbol{x}(a) \pi^*(a, \boldsymbol{X})|$$

$$< o(1)$$

The first difference is zero, as the set $\mathcal{S}$ is sampled uniformly at random. The second difference is $o(1)$ because the probability of clustering an arm incorrectly is $o(N_\mathcal{S}^{-1})$. The third difference is $o(1)$ because the regularized ordinary least squares estimator is consistent when there is no clustering error, i.e., $\hat{\boldsymbol{\mu}}_{c, N_\mathcal{S} T_0 + 1} \xrightarrow{p} \boldsymbol{\mu}_c$ as $(N_\mathcal{S}, T_0) \to \infty$ (and as long as the set $[C]$ is covered by $\mathcal{S}$), and the clustering error vanishes asymptotically. The fourth difference is $o(1)$ because it is the difference between an expectation and its empirical counterpart. $\qquad\square$

**Lemma D.3** (Formal version of Lemma 5.4). $|r(\pi') - \rho(\pi')| < o(1)$

*Proof.*

$$|r(\pi') - \rho(\pi')|$$

$$\leqslant |\mathop{\mathbb{E}}_{\boldsymbol{X}} \Big[ \sum_{a \in [K]} \boldsymbol{\mu}_{c(a)}^\top \boldsymbol{x}(a) \pi'(a, \boldsymbol{X}) \Big] - \frac{K}{N_\mathcal{S}} \mathop{\mathbb{E}}_{\boldsymbol{X}} \Big[ \sum_{a \in \mathcal{S}} \boldsymbol{\mu}_{c(a)}^\top \boldsymbol{x}(a) \pi'(a, \boldsymbol{X}) \Big]|$$

$$+ |\frac{K}{N_\mathcal{S}} \mathop{\mathbb{E}}_{\boldsymbol{X}} \Big[ \sum_{a \in \mathcal{S}} \boldsymbol{\mu}_{c(a)}^\top \boldsymbol{x}(a) \pi'(a, \boldsymbol{X}) \Big] - \frac{K}{N_\mathcal{S}} \mathop{\mathbb{E}}_{\boldsymbol{X}} \Big[ \sum_{a \in \mathcal{S}} \boldsymbol{\mu}_{\hat{c}(a)}^\top \boldsymbol{x}(a) \pi'(a, \boldsymbol{X}) \Big]|$$

$$+ |\frac{K}{N_\mathcal{S}} \mathop{\mathbb{E}}_{\boldsymbol{X}} \Big[ \sum_{a \in \mathcal{S}} \boldsymbol{\mu}_{\hat{c}(a)}^\top \boldsymbol{x}(a) \pi'(a, \boldsymbol{X}) \Big] - \frac{K}{N_\mathcal{S}} \mathop{\mathbb{E}}_{\boldsymbol{X}} \Big[ \sum_{a \in \mathcal{S}} \hat{\boldsymbol{\mu}}_{\hat{c}(a), N_\mathcal{S} T_0 + 1}^\top \boldsymbol{x}(a) \pi'(a, \boldsymbol{X}) \Big]|$$

$$+ |\frac{K}{N_\mathcal{S}} \mathop{\mathbb{E}}_{\boldsymbol{X}} \Big[ \sum_{a \in \mathcal{S}} \hat{\boldsymbol{\mu}}_{\hat{c}(a), N_\mathcal{S} T_0 + 1}^\top \boldsymbol{x}(a) \pi'(a, \boldsymbol{X}) \Big] - \frac{K}{N_\mathcal{S}} \frac{1}{N_\mathcal{S} T_0} \sum_{t=1}^{N_\mathcal{S} T_0} \sum_{a \in \mathcal{S}} \hat{\boldsymbol{\mu}}_{\hat{c}(a), N_\mathcal{S} T_0 + 1}^\top \boldsymbol{x}(a) \pi'(a, \boldsymbol{X})|$$

$$< o(1)$$

Each difference is $o(1)$ following arguments similar to those in the proof of Lemma D.2. $\qquad\square$

The result follows by considering the Karush-Kuhn-Tucker conditions to incorporate the constraints.

## D.2 USEFUL RESULTS FOR THEOREM D.8

**Lemma D.4** (Formal version of Lemma 5.6). *With probability at least* $(1-\zeta)^3$ *we have:*

*a)* $|\sum_{t=N_S T_0+1}^{T_\omega} r_t(a_t) - \boldsymbol{x}_t(a_t)^\top \widetilde{\boldsymbol{\mu}}_{a_t,t}| \leqslant R(T)$

*b)* $\|\sum_{t=N_S T_0+1}^{T_\omega} \boldsymbol{v}_t(a_t) - \boldsymbol{x}_t(a_t)^\top \widetilde{\boldsymbol{W}}_{a_t,t}\|_\infty \leqslant R(T)$

*Proof.* Considering part $a)$, we start with the probability of the event of interest, and we utilize Lemma 4.4 $c)$ in order to make the first $(1-\zeta)$ term show up, and the Azuma-Hoeffding inequality for the remaining two $(1-\zeta)$ terms.

$$\Pr\Big[|\sum_{t=N_S T_0+1}^{T_\omega} r_t(a_t) - \boldsymbol{x}_t(a_t)^\top \widetilde{\boldsymbol{\mu}}_{a_t,t}| \leqslant R(T)\Big]$$

$$= \Pr\Big[|\sum_{t=N_S T_0+1}^{T_\omega} r_t(a_t) - \boldsymbol{x}_t(a_t)^\top (\widetilde{\boldsymbol{\mu}}_{a_t,t} + \boldsymbol{\mu}_{\widehat{c}(a_t)} - \boldsymbol{\mu}_{\widehat{c}(a_t)})| \leqslant R(T)\Big]$$

$$\geqslant \Pr\Big[|\sum_{t=N_S T_0+1}^{T_\omega} r_t(a_t) - \boldsymbol{x}_t(a_t)^\top \boldsymbol{\mu}_{\widehat{c}(a_t)}| + |\sum_{t=N_S T_0+1}^{T_\omega} \boldsymbol{x}_t(a_t)^\top (\boldsymbol{\mu}_{\widehat{c}(a_t)} - \widetilde{\boldsymbol{\mu}}_{a_t,t})| \leqslant R(T)\Big]$$

$$\geqslant \Pr\Big[|\sum_{t=N_S T_0+1}^{T_\omega} r_t(a_t) - \boldsymbol{x}_t(a_t)^\top \boldsymbol{\mu}_{\widehat{c}(a_t)}| \leqslant \frac{R(T)}{2}\Big]$$

$$\cdot \Pr\Big[|\sum_{t=N_S T_0+1}^{T_\omega} \boldsymbol{x}_t(a_t)^\top (\boldsymbol{\mu}_{\widehat{c}(a_t)} - \widetilde{\boldsymbol{\mu}}_{a_t,t})| \leqslant \frac{R(T)}{2}\Big]$$

$$\geqslant \Pr\Big[|\sum_{t=N_S T_0+1}^{T_\omega} r_t(a_t) - \boldsymbol{x}_t(a_t)^\top \boldsymbol{\mu}_{\widehat{c}(a_t)}| \leqslant \frac{R(T)}{2}\Big] \cdot (1-\zeta)$$

$$= \Pr\Big[|\sum_{t=N_S T_0+1}^{T_\omega} r_t(a_t) - \boldsymbol{x}_t(a_t)^\top (\boldsymbol{\mu}_{\widehat{c}(a_t)} + \boldsymbol{\mu}_{c(a_t)} - \boldsymbol{\mu}_{c(a_t)})| \leqslant \frac{R(T)}{2}\Big] \cdot (1-\zeta)$$

$$\geqslant \Pr\Big[|\sum_{t=N_S T_0+1}^{T_\omega} r_t(a_t) - \boldsymbol{x}_t(a_t)^\top \boldsymbol{\mu}_{c(a_t)}| + |\sum_{t=N_S T_0+1}^{T_\omega} \boldsymbol{x}_t(a_t)^\top (\boldsymbol{\mu}_{c(a_t)} - \boldsymbol{\mu}_{\widehat{c}(a_t)})| \leqslant \frac{R(T)}{2}\Big]$$
$$\cdot (1-\zeta)$$

$$\geqslant \Pr\Big[|\sum_{t=N_S T_0+1}^{T_\omega} r_t(a_t) - \boldsymbol{x}_t(a_t)^\top \boldsymbol{\mu}_{c(a_t)}| + \sum_{t=N_S T_0+1}^{T_\omega} \mathbb{1}[\widehat{c}(a_t) \neq c(a_t)] \leqslant \frac{R(T)}{2}\Big] \cdot (1-\zeta)$$

$$\geqslant \Pr\Big[|\sum_{t=N_S T_0+1}^{T_\omega} r_t(a_t) - \boldsymbol{x}_t(a_t)^\top \boldsymbol{\mu}_{c(a_t)}| \leqslant \frac{R(T)}{4}\Big]$$

$$\cdot \Pr\Big[\sum_{t=N_S T_0+1}^{T_\omega} \mathbb{1}[\widehat{c}(a_t) \neq c(a_t)] \leqslant \frac{R(T)}{4}\Big] \cdot (1-\zeta)$$

$$\geqslant \Pr\Big[|\sum_{t=N_S T_0+1}^{T_\omega} r_i(a_t) - \boldsymbol{x}_t(a_t)^\top \boldsymbol{\mu}_{c(a_t)}| \leqslant \frac{R(T)}{4}\Big] \cdot \Big(1 - 2\exp\Big(-\frac{(R(T)/4)^2}{2(T_\omega - N_S T_0)}\Big)\Big)$$
$$\cdot (1-\zeta)$$

$$\geqslant \Pr\Big[|\sum_{t=N_S T_0+1}^{T_\omega} r_t(a_t) - \boldsymbol{x}_t(a_t)^\top \boldsymbol{\mu}_{c(a_t)}| \leqslant \frac{R(T)}{4}\Big] \cdot (1-\zeta)^2$$

$$\geqslant \Big(1 - 2\exp\Big(-\frac{(R(T)/4)^2}{2(T_\omega - N_S T_0)}\Big)\Big) \cdot (1-\zeta)^2$$

$$\geqslant (1 - \zeta)^3$$

The proof for part $b)$ follows the same steps. $\qquad\square$

Now, let $\mathcal{S}_1$ denote the subset of $\mathcal{S}$ that contains correctly clustered arms,

$$\mathcal{S}_1 := \{a \in \mathcal{S} : \widehat{c}(a) = c(a)\}$$

The following lemma provides a lower bound related to the choice of the algorithm.

**Lemma D.5** (Formal version of Lemma 5.7). *For $t > N_\mathcal{S} T_0$, the following inequality holds with high probability:*

$$\boldsymbol{x}_t(a_t)^\top (\widetilde{\boldsymbol{\mu}}_{a_t,t} - Z\widetilde{\boldsymbol{W}}_{a_t,t}\boldsymbol{\theta}_t) \geqslant \frac{1}{\sum_{a' \in \mathcal{S}_1} \pi^*(a', \boldsymbol{X}_t)} \sum_{a \in \mathcal{S}_1} \pi^*(a, \boldsymbol{X}_t) \cdot \boldsymbol{x}_t(a)^\top (\boldsymbol{\mu}_{c(a)} - ZW_{c(a)}\boldsymbol{\theta}_t)$$

*Proof.* Let $\Pi^\mathcal{S}$ denote the set of static policies that assign non-zero probability only to arms in $\mathcal{S}$, and let $\Pi^{\mathcal{S}_1}$ be defined equivalently. Then, for $t > N_\mathcal{S} T_0$ we have

$$
\begin{aligned}
&\boldsymbol{x}_t(a_t)^\top (\widetilde{\boldsymbol{\mu}}_{a_t,t} - Z\widetilde{\boldsymbol{W}}_{a_t,t}\boldsymbol{\theta}_t) \\
&\geqslant \max_{\pi^\mathcal{S} \in \Pi^\mathcal{S}} \sum_{a \in \mathcal{S}} \pi^\mathcal{S}(a, \boldsymbol{X}_t) \cdot \boldsymbol{x}_t(a)^\top (\widetilde{\boldsymbol{\mu}}_{a,t} - Z\widetilde{\boldsymbol{W}}_{a,t}\boldsymbol{\theta}_t) \\
&\geqslant \max_{\pi^\mathcal{S} \in \Pi^{\mathcal{S}_1}} \sum_{a \in \mathcal{S}_1} \pi^\mathcal{S}(a, \boldsymbol{X}_t) \cdot \boldsymbol{x}_t(a)^\top (\widetilde{\boldsymbol{\mu}}_{a,t} - Z\widetilde{\boldsymbol{W}}_{a,t}\boldsymbol{\theta}_t) \\
&\geqslant \frac{1}{\sum_{a' \in \mathcal{S}_1} \pi^*(a', \boldsymbol{X}_t)} \sum_{a \in \mathcal{S}_1} \pi^*(a, \boldsymbol{X}_t) \cdot \boldsymbol{x}_t(a)^\top (\widetilde{\boldsymbol{\mu}}_{a,t} - Z\widetilde{\boldsymbol{W}}_{a,t}\boldsymbol{\theta}_t) \\
&\geqslant \frac{1}{\sum_{a' \in \mathcal{S}_1} \pi^*(a', \boldsymbol{X}_t)} \sum_{a \in \mathcal{S}_1} \pi^*(a, \boldsymbol{X}_t) \cdot \boldsymbol{x}_t(a)^\top (\boldsymbol{\mu}_{\widehat{c}(a)} - ZW_{\widehat{c}(a)}\boldsymbol{\theta}_t) \\
&= \frac{1}{\sum_{a' \in \mathcal{S}_1} \pi^*(a', \boldsymbol{X}_t)} \sum_{a \in \mathcal{S}_1} \pi^*(a, \boldsymbol{X}_t) \cdot \boldsymbol{x}_t(a)^\top (\boldsymbol{\mu}_{c(a)} - ZW_{c(a)}\boldsymbol{\theta}_t)
\end{aligned}
$$

where the first inequality follows from the choice of the algorithm, the second from restricting the set of arms to $\mathcal{S}_1$, the third from considering the normalization of $\pi^*$ as a policy in $\Pi^{\mathcal{S}_1}$, the fourth from Lemma 4.4 $a)$ and $b)$, and the last equality from the definition of $\mathcal{S}_1$. $\qquad\square$

**Lemma D.6** (Formal version of Lemma 5.8). *The following inequality holds with high probability:*

$$\sum_{t=N_\mathcal{S} T_0+1}^{T_\omega} \underset{\boldsymbol{X}_t}{\mathbb{E}} \Big[ \boldsymbol{x}_t(a_t)^\top \widetilde{\boldsymbol{\mu}}_{a_t,t} \Big] \geqslant O\Big( \frac{N_\mathcal{S} \cdot T_\omega}{K \cdot T} \text{OPT}$$

$$+ Z \sum_{t=N_\mathcal{S} T_0+1}^{T_\omega} \underset{\boldsymbol{X}_t}{\mathbb{E}} \Big[ \boldsymbol{x}_t(a_t)^\top \widetilde{\boldsymbol{W}}_{a_t,t} - \frac{N_\mathcal{S}}{K} \cdot \frac{B}{T} \boldsymbol{1}_d \Big] \boldsymbol{\theta}_t \Big)$$

*Proof.* Lemma D.5 implies the weaker condition where expectation is taken over $\boldsymbol{X}_t$ only and it is conditional on the past realizations of the context:

$$
\begin{aligned}
&\underset{\boldsymbol{X}_t}{\mathbb{E}} \Big[ \boldsymbol{x}_t(a_t)^\top (\widetilde{\boldsymbol{\mu}}_{a_t,t} - Z\widetilde{\boldsymbol{W}}_{a_t,t}\boldsymbol{\theta}_t) \Big] \\
&\geqslant \underset{\boldsymbol{X}_t}{\mathbb{E}} \Big[ \frac{1}{\sum_{a' \in \mathcal{S}_1} \pi^*(a', \boldsymbol{X}_t)} \sum_{a \in \mathcal{S}_1} \pi^*(a, \boldsymbol{X}_t) \cdot \boldsymbol{x}_t(a)^\top (\boldsymbol{\mu}_{c(a)} - ZW_{c(a)}\boldsymbol{\theta}_t) \Big]
\end{aligned}
$$

$$\geq \underset{\boldsymbol{X}_t}{\mathbb{E}} \Big[ \sum_{a \in \mathcal{S}_1} \pi^*(a, \boldsymbol{X}_t) \cdot \boldsymbol{x}_t(a)^\top \big( \boldsymbol{\mu}_{c(a)} - ZW_{c(a)}\boldsymbol{\theta}_t \big) \Big]$$

$$= \underset{\boldsymbol{X}_t}{\mathbb{E}} \Big[ \sum_{a \in \mathcal{S}_1} \pi^*(a, \boldsymbol{X}_t) \cdot \boldsymbol{x}_t(a)^\top \boldsymbol{\mu}_{c(a)} \Big] - Z \underset{\boldsymbol{X}_t}{\mathbb{E}} \Big[ \sum_{a \in \mathcal{S}_1} \pi^*(a, \boldsymbol{X}_t) \cdot \boldsymbol{x}_t(a)^\top \boldsymbol{W}_{c(a)} \Big] \boldsymbol{\theta}_t$$

$$= O\Big(\frac{N_{\mathcal{S}}}{K}\Big) \cdot \Big( \frac{\text{OPT}}{T} - Z\frac{B}{T}\mathbf{1}_d\boldsymbol{\theta}_t \Big)$$

where the second step holds since the probability-normalization term is greater than one. From Lemma A.3 the size of $\mathcal{S}_1$ is $N_{\mathcal{S}}(1 - o(N_{\mathcal{S}}^{-1})) = O(N_{\mathcal{S}})$. We get the statement of the lemma by summing from period $N_{\mathcal{S}}T_0 + 1$ to period $T_\omega$. $\qquad\square$

Since in the first $N_{\mathcal{S}}T_0$ periods the choices are made randomly, and so $N_{\mathcal{S}}T_0$ of the budget can potentially be consumed, the following lemma which is known from the literature is expressed in terms of $B' = B - N_{\mathcal{S}}T_0$ and $T' = T - N_{\mathcal{S}}T_0$ rather than $B$ and $T$.

**Lemma D.7.** $\frac{N_{\mathcal{S}} \cdot B}{K \cdot T} < 2\frac{B'}{T'}$

*Proof.* $\frac{N_{\mathcal{S}}BT'}{KTB'} \leq \frac{N_{\mathcal{S}}B}{KB'} \leq \frac{B}{B'} = \frac{1}{1 - \frac{N_{\mathcal{S}}T_0}{B}} < 2$, since $B/2 > N_{\mathcal{S}}T_0$. $\qquad\square$

**Theorem D.8** (Main Result, formal version of 5.10). *For $\delta \in (0, \frac{1}{2})$, and $B > N_{\mathcal{S}}T_0$, with high probability*

$$\text{regret}(T) \leq O\Big( R(T)\big(1 + \frac{N_{\mathcal{S}}\,\text{OPT}}{KB'}\big) + \text{OPT}\big(1 - \frac{N_{\mathcal{S}}}{K}\big) \Big)$$

*where*

$$R(T) = O\Big( Cp_{\min}^{-1}m^{\frac{3}{2}}T^{1-\delta}\sqrt{\log(T)} \Big).$$

*Proof.* Let $T_\omega \leq T$ be the stopping time of the algorithm. Starting from the definition of regret we get:

$$regret(T) = \text{OPT} - \sum_{t=1}^{T} r_t(a_t)$$

$$= \text{OPT} - \sum_{t=1}^{T_\omega} r_t(a_t)$$

$$= \text{OPT} - \sum_{t=1}^{N_{\mathcal{S}}T_0} r_t(a_t) - \sum_{t=N_{\mathcal{S}}T_0+1}^{T_\omega} r_t(a_t)$$

$$\leq \text{OPT} - \sum_{t=N_{\mathcal{S}}T_0+1}^{T_\omega} r_t(a_t)$$

where the inequality follows since $r_t(a_t) \in [0, 1]$. Now, let

$$R(T) := O\Big( Cm\sqrt{T\log\big(dTm/\varsigma\big)\log(T)} + C\epsilon_c m^{\frac{3}{2}}T\sqrt{\log(T)} \Big)$$

The proof now proceeds with first stating Lemma D.4 that allows us to work with the optimistic estimates instead of the actual realizations of the reward and the consumption. $\qquad\square$

Thus, we have

$$\sum_{t=N_{\mathcal{S}}T_0+1}^{T_\omega} \underset{\boldsymbol{X}_t}{\mathbb{E}} \Big[ \boldsymbol{x}_t(a_t)^\top \tilde{\boldsymbol{\mu}}_{a_t,t} \Big]$$

$$\geqslant O\Big(\frac{N_{\mathcal{S}}T_\omega}{KT}\operatorname{OPT}+Z\sum_{t=N_{\mathcal{S}}T_0+1}^{T_\omega}\underset{\boldsymbol{X}_t}{\mathbb{E}}\Big[\boldsymbol{x}_t(a_t)^\top\widetilde{\boldsymbol{W}}_{a_t,t}-2\frac{B'}{T'}\boldsymbol{1}_d\Big]\boldsymbol{\theta}_t\Big) \tag{19}$$

$$\geqslant O\Big(\frac{N_{\mathcal{S}}T_\omega}{KT}\operatorname{OPT}+Z\Big(2B'\big(1-\frac{T_\omega-N_{\mathcal{S}}T_0}{T'}\big)-R(T)\Big)\Big)$$

$$\geqslant O\Big(\frac{N_{\mathcal{S}}T_\omega}{KT}\operatorname{OPT}+\frac{N_{\mathcal{S}}\operatorname{OPT}}{2KB'}\Big(2B'\big(1-\frac{T_\omega-N_{\mathcal{S}}T_0}{T'}\big)-R(T)\Big)\Big)$$

$$\geqslant O\Big(\frac{N_{\mathcal{S}}}{K}\operatorname{OPT}\Big(\frac{T_\omega}{T}+1-\frac{T_\omega-N_{\mathcal{S}}T_0}{T'}-\frac{R(T)}{2B'}\Big)\Big)$$

$$\geqslant O\Big(\frac{N_{\mathcal{S}}}{K}\operatorname{OPT}\Big(1-\frac{R(T)}{B'}\Big)\Big) \tag{20}$$

where the first step follows from Lemmas D.6 and D.7, the second from Lemma 5.9, and the third from Lemma 5.5.

From Lemma D.4, the bound in Eq. (20) applies to $\sum_{t=N_{\mathcal{S}}T_0+1}^{T_\omega}\mathbb{E}_{\boldsymbol{X}_t}\big[r_t(a_t)\big]$ by adding $R(T)$, and an application of the Azuma-Hoeffding inequality to the realized reward $\sum_{t=N_{\mathcal{S}}T_0+1}^{T_\omega}r_t(a_t)$ then gives:

$$regret(T)\leqslant O\Big(\operatorname{OPT}-\frac{N_{\mathcal{S}}}{K}\operatorname{OPT}\big(1-\frac{R(T)}{B'}\big)+R(T)\Big)$$

$$=O\Big(\operatorname{OPT}\big(1-\frac{N_{\mathcal{S}}}{K}\big)+R(T)\big(1+\frac{N_{\mathcal{S}}\operatorname{OPT}}{KB'}\big)\Big).$$

## LLM USAGE DISCLOSURE

LLMs were used only to polish language, such as grammar and wording. These models did not contribute to idea creation or writing, and the authors take full responsibility for this paper's content.

