# OpenReview forum: "One-Shot Clustering for Contextual Bandits with Knapsacks"
_ICLR.cc/2026/Conference — ICLR 2026 Conference Withdrawn Submission_

### Official Review · Reviewer_QqcZ · 2025-10-25

**Soundness:** 3
**Presentation:** 2
**Contribution:** 3
**Rating:** 2
**Confidence:** 4

**Summary:**

This work studies the contextual bandits problem with knapsack constraints, with the additional condition that the total $K$ arms are grouped into $C$ clusters, and only the arms in a pre-determined subset $\mathcal S \subset [N]$ can be played.
Meanwhile, contexts are i.i.d. across arms and timeslots.
In this problem, this paper proposes an algorithm that is based on OFU (optimism in the face of uncertainty), the algorithm achieves a regret of $O(T^{1 - \delta})$ when the resource is $\tilde{\Omega}(T^{2\delta})$ for $\delta \in (0, 1/2)$.

**Strengths:**

On the pro side, I think the problem is well motivated, and the model is appropriate.
The solution of this paper combines the methods from clustering and contextual bandits with knapsacks, and leads to a no-regret result, which is generally satisfying.

**Weaknesses:**

The authors do not discuss the widely-studied case that $\delta = 1/2$, or that the resource constraint increases linearly with the timespan.
This is an important case that is covered by works in the CBwK literature.
(Btw, in Line 71, $B > \tilde{O}(T^{2\delta})$ is not an appropriate expression.
It seems that you mean $B = \tilde{\Omega}(T^{2\delta})$?)

Meanwhile, I highly suggest that the authors improve their organization and writing of this paper.
For example, in Section 4, the paper keeps using $N_{\mathcal S}$ and $T_0$ before they are formally defined in Section 5.
In Section 5, $\rho(\pi)$ is also used before being defined.
Also, the clustering process is almost not discussed in the main body, only by a sentence in Line 322 and in Line 5 of Algorithm 1.
This procedure should be emphasized more since it is an important part of this work, yet the detailed clustering method is only given in the Appendix.
I encourage the authors to talk more about intuitions on the clustering in the main body and put less effort on the technical lemmas given by the clustering.
This will lessen the burden of reading the paper.

Further, some folklores are not mentioned in this paper, e.g., $OPT$ gives an upper bound on all online algorithms.
As another example, the online mirror descent algorithm, though widely used, is not given in the paper, which makes the paper a bit incomplete.
Meanwhile, the citation could be more appropriate.
For example, no citation is given in the main body when the confidence ellipsoid is discussed, yet this is a common technique in the online learning literature.

In fact, I have reviewed this paper in NeurIPS 2025, when the authors committed to having ALREADY resolved the above issues in the manuscript.
I failed to see that, unfortunately.
I have to give an even lower score based on this.

**Questions:**

See the above "weakness" part.

---

> ### Author Response · Authors · 2025-12-01
> **Thank you for your review**
>
> The authors sincerely thank the reviewers for their time and thoughtful comments, and we appreciate the feedback provided.

---

### Official Review · Reviewer_axvH · 2025-10-28

**Soundness:** 3
**Presentation:** 2
**Contribution:** 2
**Rating:** 4
**Confidence:** 3

**Summary:**

This paper studies contextual bandits with knapsack constraints in which arms share unknown cluster structures. It proposes clusterLCBwK, an algorithm that performs one-time clustering on a small subset of sampled arms using Classifier-Lasso, then optimizes decisions under resource constraints based on estimated cluster parameters. The algorithm jointly learns the cluster structure and the linear models, achieving high-probability cluster identification and sublinear regret bounds.

**Strengths:**

- The paper provides a thorough theoretical analysis of the proposed algorithm, including regret bounds and cluster identification errors.
- The proposed algorithm combines techniques for clustering using Classifier-Lasso with techniques for bandits with budget constraints.

**Weaknesses:**

- The assumptions that the number of clusters is known and fixed, and the smallest cluster proportion $p_{\min}$ is known, are restrictive and may not hold in real-world scenarios.
- No empirical evaluation is provided to validate the theoretical findings and demonstrate the algorithm's performance in practice.

**Questions:**

- Could the authors discuss the practical applicability of the assumption that the number of clusters is known and fixed, or is there any solution to relax this assumption?
- Comparing to LinCBwK, it seems that the dependence on the context dimension $m$ is worse in clusterLCBwK. Could the authors discuss this?

---

> ### Author Response · Authors · 2025-12-01
> **Thank you for your review**
>
> The authors sincerely thank the reviewers for their time and thoughtful comments, and we appreciate the feedback provided.

---

### Official Review · Reviewer_5n9t · 2025-11-01

**Soundness:** 3
**Presentation:** 2
**Contribution:** 2
**Rating:** 4
**Confidence:** 3

**Summary:**

This paper studies the linear contextual bandit problem with knapsack constraints, motivated by real-world scenarios that involve resource and fairness considerations. To address these challenges, the authors propose an algorithm that clusters actions to enable knowledge transfer across similar items. The paper provides a formal regret analysis showing that the proposed algorithm achieves sublinear regret in $T$. A notable feature is that clustering only needs to be performed once on a randomly selected subset of actions.

**Strengths:**

1. The problem is well-motivated and captures realistic constraints such as fairness and limited resources. The combination of arm clustering and knapsack constraints is natural and timely.
2. The theoretical contribution is solid and helps to tackle an existing gap in the literature.
3. The discussion on the dependence on $K$ is appreciated and provides useful insight.

**Weaknesses:**

1. The paper lacks experimental validation. Including even synthetic experiments would help demonstrate the practical effectiveness of the proposed one-shot clustering approach and the impact of $K$.
2. Key assumptions should be stated clearly in the main text and discussed in terms of their realism or applicability to practical settings.
3. Some references appear misplaced or unclear. For example, in line 128, *Wang et al. (2024)* and *Yang et al. (2024)* seem to belong to the discussion in line 120. Additionally, the phrase “requiring a lower bound on the number of units to be clustered” (line 129) is vague and should be clarified.

**Questions:**

Please refer to the weakness part above.

---

> ### Author Response · Authors · 2025-12-01
> **Thank you for your review**
>
> The authors sincerely thank the reviewers for their time and thoughtful comments, and we appreciate the feedback provided.

---

### Official Review · Reviewer_B6Bg · 2025-11-01

**Soundness:** 3
**Presentation:** 2
**Contribution:** 2
**Rating:** 4
**Confidence:** 3

**Summary:**

This paper studies clustered contextual bandits with knapsack constraints, where arms belong to unknown clusters (but the number of clusters $C$ is assumed known). Arms in the same cluster share linear reward & consumption models. The authors propose clusterLCBwK. The algorithm first performs a one-shot clustering on a randomly sampled subset of arms, then runs a conventional optimism-in-the-face-of-uncertainty policy over the subset. The authors prove that this algorithm achieves sublinear regret of order $T^{1-\delta}$ for $\delta \in (0,1/2)$ when the resource budgets scale as $T^{2\delta}$.

**Strengths:**

1. To my knowledge, this is the first work that integrates clustered contextual bandits and bandits with knapsacks.
2. The algorithm design is simple and intuitive. It discovers cluster structure once, then leverages the standard bandit with knapsacks machinery.
3. The authors provide a strong theoretical analysis. The regret bound clearly exposes how performance depends on subset size, cluster coverage, budget scaling, and clustering error.

**Weaknesses:**

1. The analysis requires heavy assumptions such as i.i.d. contexts, known number of clusters $C$, and knowledge of or lower bounds on parameters such as $p_{\min}$. In many applications, these are unknown.
2. In the regret analysis, the dependence on $m$ (feature dimension) is $m^{3/2}$, which is worse than the $m$ dependnece in non-clustered knapsacks. However, the paper does not provide a principled explanation or any ideas to close this gap.
3. The writing needs improvement. For example, one novelty of this paper is arm clustering, but the key elements of the clustering step, including its assumptions, estimator, and how mis-clustering propagates, are scattered across sections and the appendix. Centralizing these would greatly improve readability.
4. The "informal" lemmas are identical to their "formal" versions, providing no additional intuition.
5. The paper is purely theoretical. There is no empirical validation. Even small synthetic studies would be valuable.

**Questions:**

1. Can similar theoretical guarantees be achieved without knowing $C$? For example, (Gentile et al., 2014) does not require knowledge of the number of clusters.
2. Is the extra $\sqrt{m}$ factor in the regret bound intrinsic to your mis-clustering handling? Can you close this gap, or can you provide a lower bound showing that this extra factor is unavoidable?
3. How sensitive is clusterLCBwK to non-i.i.d. contexts or model misspecification?

---

> ### Author Response · Authors · 2025-12-01
> **Thank you for your review**
>
> The authors sincerely thank the reviewers for their time and thoughtful comments, and we appreciate the feedback provided.

---

### Note · Authors · 2025-12-01

**Comment:**

The authors sincerely thank the reviewers for their time and thoughtful comments, and we appreciate the feedback provided.

**Withdrawal Confirmation:**

I have read and agree with the venue's withdrawal policy on behalf of myself and my co-authors.